# Multi-drug resistant (MDR) Gram-negative pathogenic bacteria isolated from poultry in the Noakhali region of Bangladesh

**Md. Adnan Munim**[1], **Shuvo Chandra Das**[1], **Md. Murad Hossain**[1], **Ithmam Hami**[1], **Mridul Gope Topu**[2], **Shipan Das Gupta**[1]*

**1** Department of Biotechnology and Genetic Engineering, Noakhali Science and Technology University, Noakhali, Bangladesh, **2** Department of Microbiology, Noakhali Science and Technology University, Noakhali, Bangladesh

* shipan.bge@nstu.edu.bd

**Data Availability Statement:** All relevant data are within the manuscript and its Supporting Information files.

## Abstract

Rapidly increasing antibiotic-resistant bacterial strains in Bangladesh's food and farm animals stem from the excessive and inappropriate use of antibiotics. To assess the prevalence of multi-drug resistant (MDR) Gram-negative bacteria in poultry chicks, we sought to isolate and identify strains carrying antimicrobial resistance genes. Isolation and identification involved biochemical tests, *16S* rRNA sequencing, and PCR screening of species-specific genes. MDR patterns were evaluated using CLSI guidelines with seventeen antibiotics across twelve classes. Targeted gene sequences were amplified for the detection of Extended-spectrum β-Lactamase (ESBL), carbapenem, tetracycline, sulfonamide, and colistin resistance genes. Common isolates, such as *Escherichia coli*, *Klebsiella pneumoniae*, *Proteus penneri*, and *Enterobacter hormaechei*, exhibited average Multiple Antimicrobial Resistance (MAR) indices of 0.66, 0.76, 0.8, 0.84, and 0.81, 0.76, 0.84, 0.41 for broiler and layer chicken, respectively. *Providencia stuartii* and *Salmonella enterica*, exclusive to broiler samples, had MAR indices of 0.82 and 0.84, respectively. Additional isolates *Morganella morganii*, *Aeromonas* spp., and *Wohlfahrtiimonas chitiniclastica* were found in layers (Average MAR indices: 0.73, 0.71, and 0.91). Notably, *M. morganii*, *E. hormaechei and W. chitiniclastica* were identified for the first time in Bangladeshi poultry chicken, although their evolution is yet to be understood. In this study, Pan-drug resistance was observed in one *P. stuartii* (broiler) and one *Aeromonas* spp. (layer) with a MAR index 1, while all isolates exhibited MAR indices >0.2, indicating MDR. Antimicrobial resistance (AMR) gene screening identified *bla*TEM, *bla*SHV, *tetA*, and *sul1* in a majority of the MDR strains. Interestingly, *E. coli* (lactose positive and negative) and *E. hormaechei* were exclusively found to possess the *tetB* gene. In addition, *E. coli* (lactose negative), *Klebsiella pneumoniae*, *Enterobacter hormaechei*, *M. morganii*, and *P. stuartii* were observed to carry the colistin-resistant *mcr-1* gene, whereas *sul2* was detected in *E. coli* (lactose positive and negative), *E. hormaechei*, *P. stuartii*, and *P. penneri*. These findings emphasize the health risk of our consumers of both broiler and layer chickens as they have turned into a potent reservoir of various AMR gene carrying MDR and Pan-drug resistant bacteria.

**Funding:** This work was supported by the 'Noakhali Science and Technology University- Research Cell' Teachers' grant of the budget year 2021-2022 (Project ID: NSTU/RC-BG-06/T-23/32). The funders contributed solely by providing financial support for this study. They were not involved in the study's design, data collection, analysis, the decision to publish, or the preparation of the manuscript.

**Competing interests:** The authors have declared that no competing interests exist.

## Introduction

More recently, antibiotic resistant (AR) bacteria are considered one of the biggest threats to global health, food security, and development [1]. Bacteria are becoming stronger day by day against diverse group of antibiotics [2, 3]. The impact of antibiotic resistance is indiscriminate, posing a threat to individuals of all ages, regardless of geographical location. As a consequence, a growing list of infections–such as pneumonia, tuberculosis, blood poisoning, gonorrhea, and foodborne diseases–are becoming harder, and sometimes impossible, to treat as antibiotics become less effective [4]. Antibiotics are used for therapeutic purposes and disease prevention in veterinary medicine and are commonly added to animal feed at sub-therapeutic levels for growth promotion. According to reports, Asia consumed the largest quantity of antibiotics, totaling approximately 57,167 tonnes for non-therapeutic purposes in 2017. Projections indicate that this figure is expected to rise to 63,062 tonnes by the year 2030 [5]. With global consumption of chicken as a protein source and the projected rapid growth of agro-based poultry industries, this trend is set to intensify [6]. According to the Department of Livestock Services 2020–21, there are more than 350 million poultry in Bangladesh which contributes to the growth of the national economy and the creation of job opportunities. Furthermore, many people benefit from the poultry industry since it provides a cheaper and more conveniently accessible source of nutrition and protein in the form of eggs and meat [7, 8]. Due to the concern about the care and maintenance of chicken health, poultry breeders use antibiotics at therapeutic and non-therapeutic levels to avoid disease and improve feed utilization and growth performance to meet consumer demand [6].

The *Enterobacteriaceae* family comprises a diverse group of Gram-negative bacteria found in water, decaying waste, soil, and the gastrointestinal tracts of both humans and animals. These bacteria are often responsible for diarrhea, primarily transmitted through contaminated food or water [9]. Notably, pathogenic bacteria like *Salmonella spp.*, *Escherichia coli*, *Yersinia spp.*, *Klebsiella species*, and *Shigella spp.* are commonly linked to gastrointestinal complications in humans and are frequently identified in chickens and their associated food products [10–13]. *Enterobacteriaceae* frequently serve as carriers of extended-spectrum β-lactamase (ESBL)-encoding genes and are frequently found in poultry and their surrounding environment, contributing to the prevalence of ESBL-producing strains [14]. The predominant cause of antibiotic resistance in *Enterobacteriaceae* is the production of ESBLs, AmpC-lactamases, and carbapenemases, rendering them resistant to a diverse array of antibiotics. Apart from penicillins and first, second, and third generation cephalosporins, these bacteria exhibit resistance to multiple other antimicrobials. Notably, tetracycline, fluoroquinolone, and trimethoprim resistance have rapidly spread worldwide, driven by the proliferation of ESBL genes [15, 16]. The presence of *bla*CTX-M, *bla*SHV, and *bla*TEM genes encoding CTX-M, TEM, and SHV β-lactamases, respectively, empowers these bacteria with resistance against penicillins, first, second, and third-generation cephalosporins, as well as aztreonam [17, 18]. Notably, studies by Kluytmans et al. [19] and Leverstein-van Hall et al. [20] have highlighted strong genetic similarities between ESBL-producing *E. coli* isolated from chicken meat and humans. The identification of CTX-M-1 and TEM-52 genetic traits on similar plasmids in *E. coli* from these distinct sources further supports the potential transmission of ESBL genes through food pathways [19, 20]. The presence of diverse ESBL and carbapenemase genetic elements, along with numerous other antibiotic resistance determinants, on mobile genetic elements presents an ongoing challenge. This scenario has the potential to lead to the emergence of bacteria resistant to all available antibacterial agents [21–23]. Colistin (polymyxin E) usage for therapeutic, preventative, and growth promotion purposes in food animals has been prevalent in several Asian countries, including India, Japan, Korea, and Vietnam. In Bangladesh, colistin is already allowed as a

veterinary therapeutic agent and a dietary supplement, which is used to treat infections caused by multi-drug resistant Gram-negative microbes as a penultimate resource [24–27]. Moreover, the mobilization of the colistin resistance gene variant-1 (*mcr-1*) represents a plasmid-mediated mechanism, and additional variants of the mcr gene (*mcr-1* to *mcr-9*) have been detected in *Enterobacteriaceae* species across various countries, including Bangladesh. These findings indicate that these genes are present in natural environments, livestock, and humans, posing a significant threat to public health [28–32]. Following this, a subsequent epidemic of colistin-resistant bacteria emerged among humans in China, resulting in a notable increase in mortality. Colistin-resistant *Enterobacteriaceae* were identified as human pathogens, in line with earlier reports [27, 33–35].

Bangladesh, despite being a developing economy, features a spectrum of poultry and food-animal farming systems that span from traditional family farms to medium- and large-scale commercial enterprises. The absence of an effective government-led animal healthcare system has led farm proprietors to depend on unskilled and informal healthcare practitioners for animal treatment. Consequently, the prevalent misuse, overuse, and suboptimal administration of antibiotics on farms have been exacerbated by unrestricted access to these drugs [36, 37]. As a result, MDR bacteria are persistently emerging within the poultry industry, and the characteristics of these MDR strains are continually changing each year. Consequently, it has become imperative to assess the present situation of MDR bacteria in the poultry sector to implement appropriate precautionary measures. For the reasons mentioned above, this study aims to isolate and characterize multi-drug resistant (MDR) bacteria from the gut and rectal swabs of broiler and layer chickens raised in poultry farms within the Noakhali region. Additionally, the study seeks to detect resistance genes (ESBL, carbapenem, tetracycline, trimethoprim, colistin) within these isolated bacteria. A comparative analysis was conducted between MDR bacteria from broiler and layer chickens, with a focus on identifying potential reasons for any observed differences. Ultimately, this research endeavor aims to enhance our understanding of multi-drug resistant bacterial species and their resistance patterns against a diverse range of antibiotics.

## Material and methods

### Ethical approval and area of study

This study received approval from the Ethics and Research Review Committee of Noakhali Science and Technology University Faculty of Sciences (Approval No. NSTU/SCI/EC/2023/186). All methods adhered to guidelines and regulations. We obtained farm and chicken information with informed consent from farm owners, ensuring their privacy and commercial data protection. We utilized sample codes during data collection to maintain accuracy and protect privacy.

We chose three distinct small-scale commercial broiler and layer farms for this study situated in Noakhali Sadar. Information collected included farm size, total chicken population, age and weight of chickens, feeding habits, disease prevalence, medications administered for treatment, and the prevalent antibiotics used within these farms (**S1** and **S2** **Tables**).

### Selection and criteria of the study site

Samples were collected from local small-scale commercial poultry farms in Noakhali Sadar upazila. Details regarding the farms and the collected samples can be found in the **S1** and **S2** **Tables**. The farms, run by 4–6 workers, handle tasks such as feeding, medication, and overall chicken care. These chickens are sold in nearby markets like Sonapur Bazar, Maijdi Pouro Bazar, and Maijdi bazar. Nearby residents often purchase eggs and poultry directly from these

farms. For our study, we gathered fresh stool and rectal swab samples from Mousumi Poultry and Nur-hossain Agro (two broiler and two layer chickens each), and from Shohag Poultry, Poultry Farms, and Dipto Poultry (one broiler and one layer chicken each).

## Sample collection

We conducted this study by procuring rectal swab and stool samples from three distinct farms situated in Noakhali Sadar. To achieve this, we selected two chickens from each farm for the collection of rectal swab and stool samples. The rectal swab specimens were meticulously collected using sterile cotton buds and were then placed in sterile falcon tubes. Stool samples, on the other hand, were gathered using sterile forceps and deposited into sterile petri plates. These Falcon tubes and petri plates were airtight sealed using Parafilm and securely stored in separate zipper bags at 4 ˚C, ensuring protection against contamination from collection to laboratory transportation. Stringent safety protocols were upheld throughout the sample collection process, and visits were limited to one farm per day to prevent any potential cross-contamination.

## Isolation and presumptive identification of bacteria

Gram-negative bacterial isolation was accomplished using the standard serial dilution plate technique, slightly modified from Bushen *et. al.* (2021) [38]. Initially, 1 g of stool sample or a rectal swab cotton bud was introduced into 9 ml of peptone water broth media and incubated at 37˚C for 18 hours. Subsequently, 1 ml of peptone water broth with the sample was transferred into 9 ml of sterile 0.9% saline water and thoroughly mixed by vortex agitation. Serial dilutions were performed, spanning up to a $10^{-5}$-fold reduction. A 0.1 ml inoculum was extracted from each dilution and spread across the surfaces of MacConkey agar, Eosin methylene blue (EMB) agar, and Xylose Lysine Deoxychocolate (XLD) agar plates. Incubation of the plates occurred at 37˚C for 24 hours. Following incubation, individual colonies were selected and streaked onto a MacConkey agar plate to achieve pure cultures. Subsequently, each bacterial isolate from the pure culture was subjected to presumptive identification using a series of tests, including the Lactose test, Triple sugar iron test, Motility test, Urease test, Indole test, and Oxidase test, in accordance with Bergey's Manual of Determinative Bacteriology [39].

## Molecular identification of bacterial isolates

Bacterial DNA extraction was conducted using the boiling method, as per [40] with some modifications. The complete extraction procedure consisted of the following steps: i) Transfer 1 ml of pre-enrichment culture (peptone water) into a sterile microcentrifuge tube. ii) Centrifuge at 16,000 g for 15 minutes and carefully remove the supernatant. iii) Repeat steps (i) and (ii). iv) Resuspend the pellet in 400 µl of DNase and RNase free water through vortexing. v) Centrifuge at 16,000 g for 10 minutes and discard the supernatant. vi) Resuspend the pellet in 200 µl of DNase-RNase free water through vortexing. vii) Incubate at 100˚C for 15 minutes and promptly chill on ice for 10 minutes. viii) Centrifuge for 5 minutes at 16,000 g at 4˚C. ix) Carefully transfer the supernatant to a new microcentrifuge tube. x) Use an aliquot of 2–5 µl of the supernatant as the template DNA for PCR.

Following the bacterial DNA extraction, presumptively identified isolates of *Klebsiella spp*. and *Aeromonas spp*. underwent PCR screening for molecular level identification. Specifically, the PCR screening for *K. pneumoniae* identification targeted the *rcsA* gene [41], while for *Aeromonas spp*. identification, the *gyrB* gene was the focus of PCR screening [42]. **S3**–**S5 Tables** outlines Primers details, PCR mixture preparation and conditions for *K. pneumoniae* and *Aeromonas spp*. detection. Further, *16S* rRNA gene amplification was performed on

presumptively identified bacteria using the 27F (5′-AGAGTTTGATCCTGGCTCAG-3′) primer as forward and the 1492R (5′-GGTTACCTTGTTACGACTT-3′) primer as reverse. Primer details, PCR mixture preparation and PCR conditions are provided in **S3, S6 and S7 Tables**. Amplified *16S* rRNA products were purified and sequenced commercially at the National Institute of Biotechnology using Sanger di-deoxy sequencing. Sequenced DNA data were analyzed with SnapGene Viewer 6.0.5 (https://www.snapgene.com/snapgene-viewer) and subjected to individual Basic Local Alignment Search Tool (BLAST) analysis against the National Center for Biotechnology Information (NCBI) blastn server for database comparison (https://blast.ncbi.nlm.nih.gov/Blast.cgi?PROGRAM=blastn&BLAST_SPEC=GeoBlast &PAGE_TYPE=BlastSearch). In *16S* rRNA sequencing, two bacterial isolates per genus were selected, validating biochemical test outcomes.

## Phylogenetic analysis of *16S* rRNA gene sequenced DNA data

Phylogenetic analysis of the bacterial isolates identified through *16S* rRNA gene sequencing was conducted using the online web tool MAFFT version 7 (https://mafft.cbrc.jp/alignment/ server/phylogeny.html) [43, 44]. For DNA analysis, a scoring matrix of 1PAM/K = 2 was employed, alongside default settings for other parameters.

## Detection of phenotypic antibiotic resistance

Phenotypic antibiotic resistance profiling of identified bacterial isolates was done by Kirby-Bauer disk-diffusion method according to Hudzicki, (2009) [45]. Bacterial isolates identified through *16S* rRNA gene sequencing, as well as those exhibiting similar biochemical profiles, were selected for antibiotic resistance testing. A panel of seventeen antibiotics from eight classes, recommended by the Center for Disease Prevention and Control (CDC) for *Enterobacteriaceae* family bacterial infections was utilized. These antibiotics encompassed Ampicillin (AMP25), Amoxicillin-clavulanic acid (AMC 30), Cefotaxime (CTX 30), Cefoxitin (CX 30), Aztreonam and Imipenem from β-lactams; Ciprofloxacin (CIP 5) and Norfloxacin (NX 10) from fluoroquinolones; Gentamicin (GEN 10) and Kanamycin (K 30) from aminoglycosides; Azithromycin (AZM 30) and Erythromycin (E 10) from macrolides; Chloramphenicol (C 30) from phenicol; Trimethoprim-Sulfamethoxazole (COT 25) from sulphonamides; Tetracycline (TE 30) from tetracyclines; Colistin (CL 10) and Polymyxin B (PB 300) from polymyxins (Himedia, India). Zone of inhibition was measured in millimeters (mm) and interpreted as sensitive (S), intermediate (I), or resistant (R) according to Clinical and Laboratory Standards Institute (CLSI) standards [46]. For colistin resistance analysis, we followed the studies by Uwizeyimana *et al* (2020) and Fadare *et. al.* (2021) [47, 48]. Additionally, a zone of inhibition <12 mm considered as resistance.

## Evaluation of Multiple antimicrobial resistance phenotype (MARP) and multiple antimicrobial resistance index (MARI)

Bacterial species demonstrating resistance to one or more antibiotics from distinct antibiotic classes are categorized as exhibiting multi-drug resistance (MDR) [49]. In accordance with this principle, bacterial isolates in this study manifesting resistance across three antibiotic classes are classified as MDR. Furthermore, the Multiple Antibiotic Resistance Index (MARI) of each isolate, an assessment of antimicrobial resistance, was calculated using an equation elucidated by [44], as follows:

$$\text{MAR index} = A_R/A_U$$

Here, "$A_R$" signifies the cumulative count of antibiotics to which bacterial isolates displayed resistance, while "$A_U$" denotes the total count of antibiotics employed. If the MARI value of any isolate surpasses 0.2, it is categorized as demonstrating multi-drug resistance (MDR) [48, 50, 51].

### Genotypic identification of antimicrobial resistant genes (ARG)

Subsequent to the identification of bacterial isolates and their patterns of multi-drug resistance, we conducted a comprehensive screening for nine distinct antibiotic resistance genes (ARGs). These encompassed Extended Spectrum β-lactamases (ESBLs) genes including *bla*CTX-M, *bla*TEM, and *bla*SHV, alongside the New Delhi Metallo β-lactamase *(bla*NDM), a metallo-carbapenemase gene. Additionally, non-β-lactamase genes such as *tetA* and *tetB* were targeted for tetracycline resistance assessment, while *sul1* and *sul2* were examined for Sulfonamides resistance. Moreover, among the globally described ten *mcr* variants encoding colistin resistance, only *mcr-1* gene was investigated by PCR since it is more prevalent worldwide. The selection criteria for ARG screening encompassed bacterial isolates demonstrating resistance or intermediate phenotypes to the aforementioned antibiotics. Primers for the ARGs mentioned above were chosen from prior studies listed in **S3 Table** and were synthesized commercially. The annealing temperature was determined using the web-based Tm calculator tool (https://tmcalculator.neb.com/#!/main). Detailed information regarding PCR mixture preparation and PCR conditions for ARG screening is provided in **S8**–**S12** **Tables**.

## Result

### Isolation and identification of bacterial isolates

A total of 12 samples were examined, including 6 stool samples each from broiler chickens and layer chickens, as well as 6 rectal swab samples (3 from broiler chickens and 3 from layer chickens). These samples were cultured on MacConkey agar, EMB agar, and XLD agar plates. A total of 160 bacterial isolates were identified from the twelve samples. The occurrence and distribution of these isolated bacterial strains in both broiler and layer chickens are presented in Fig 1.

The isolated bacteria underwent presumptive identification through conventional microbiological and biochemical techniques. As a result, nine distinct bacterial genera emerged from the pool of 160 isolates. Among these, *E. coli*, *Klebsiella spp.*, *Salmonella spp.*, *Providencia spp.*, *Enterobacter spp.*, *Proteus spp.*, and *Morganella spp.* were identified within the family *Enterobacteriaceae*. Additionally, *Aeromonas spp.* from the Aeromonadaceae family and *Pseudomonas spp.* from the Pseudomonadaceae family constituted the remaining two. Following this preliminary identification, *K. pneumoniae* and *Aeromonas spp.* were pinpointed to the species level via housekeeping gene amplification (refer to **Fig 2**), while *16S* rRNA sequencing was employed to validate the species-level identification of other bacterial isolates.

Consequently, *16S* rRNA sequencing confirmed the species identities of *E. coli*, *K. pneumoniae*, *Salmonella enterica*, *Providencia stuartii*, *Enterobacter hormaechei*, *Proteus penneri*, and *Morganella morganii* (**S1 File**). Surprisingly, our *16S* rRNA gene sequencing revealed a strain of *Wohlfahrtiimonas chitiniclastica*, which had initially been presumptively identified as *Pseudomonas spp.* through conventional microbiological and biochemical techniques. A detailed breakdown of the identity match percentages and query coverage, as compared to the NCBI database, can be found in **Table 1**.

During our investigation, we identified lactose-negative *E. coli* isolates through *16S* rRNA sequencing, leading us to hypothesize that these non-lactose fermenters belonged to bacterial groups outside the *Enterobacteriaceae* family. Notably, the culture characteristics of *E. coli* on

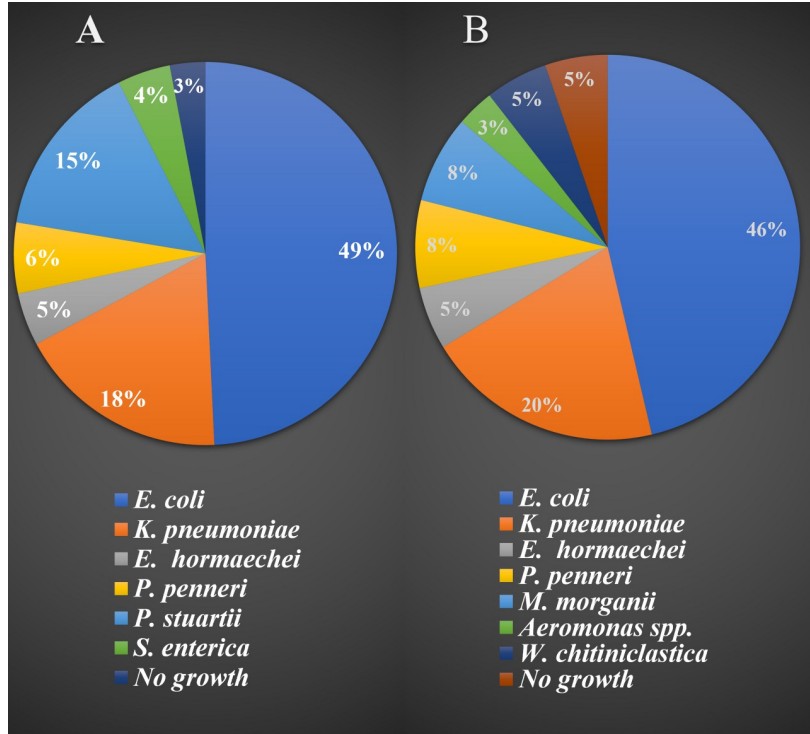

**Fig 1. Identified bacterial isolates from broiler and layer chicken stool and rectal swab.** (A represents the bacterial isolates from broiler chicken and B represents the bacterial isolates from layer chickens).

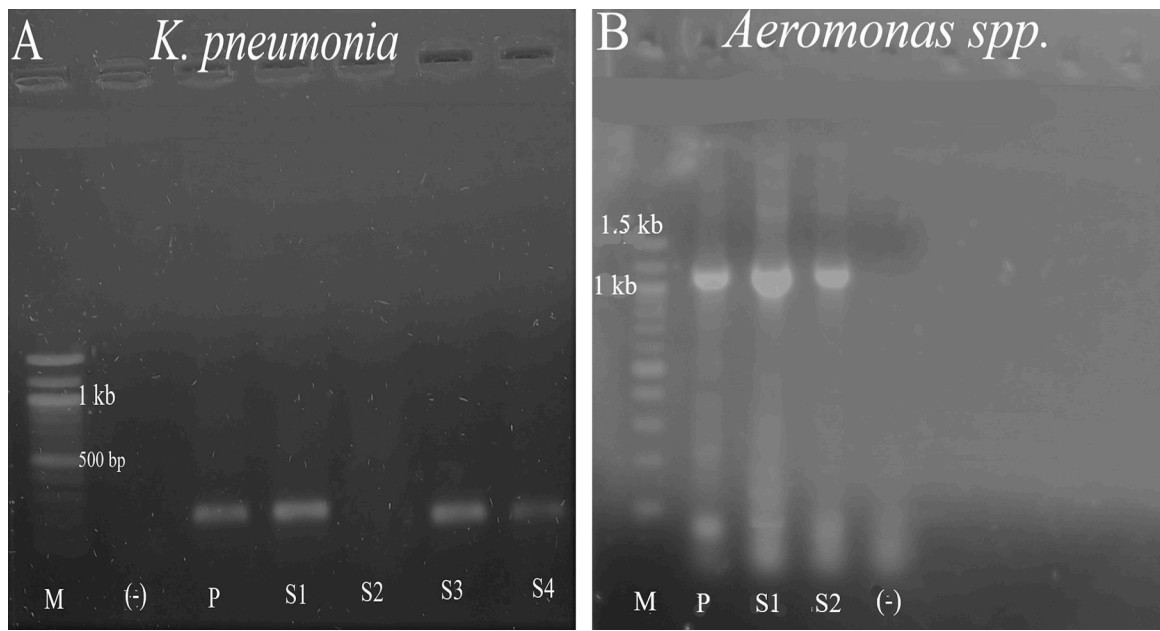

**Fig 2.** Detection of *K. pneumoniae* (A) and *Aeromonas spp.* (B) by PCR amplification of *rcsA* and *gyr-B* gene targeted segment respectively. (M = 100 bp ladder, (-) = negative control, P = positive control, S = Sample).

**Table 1.** *16S rRNA sequencing result of bacterial isolates.*

| Isolates Name | Query coverage % | Identity in percentage | Accession length in base pair | Bacterial family |
|---|---|---|---|---|
| *E. coli* (Lactose positive) | 94 | 98 | 1439 | *Enterobacteriaceae* |
| *E. coli* (Lactose negative) | 100 | 100 | 1443 | *Enterobacteriaceae* |
| *K. pneumoniae* | 100 | 99.40 | 1372 | *Enterobacteriaceae* |
| *P. stuartii* | 100 | 99.64 | 1545 | *Enterobacteriaceae* |
| *S. enterica* | 100 | 99.88 | 1069 | *Enterobacteriaceae* |
| *E. hormaechei* | 100 | 99.38 | 1196 | *Enterobacteriaceae* |
| *P. penneri* | 99 | 98.80 | 1365 | *Enterobacteriaceae* |
| *M. morganii* | 99 | 99.06 | 1464 | *Enterobacteriaceae* |
| *W. chitiniclastica* | 100 | 98.80 | 1423 | *Incertae sedis* |

MacConkey agar exhibited variation in response to lactose utilization, as illustrated in **S1 Fig**. Among the 160 bacterial isolates, *E. coli* (33 isolates from broiler and 44 isolates from layer), *K. pneumoniae* (12 isolates from broiler and 19 isolates from layer), *E. hormaechei* (3 isolates from broiler and 5 isolates from layer), and *P. penneri* (4 isolates from broiler and 7 isolates from layer) were commonly encountered in both broiler and layer chickens respectively. Conversely, *P. stuartii* (10 isolates) and *S. enterica* (4 isolates) were present solely in broiler chicken samples, while *M. morganii* (7 isolates), *Aeromonas spp.* (3 isolates), and *W. chitiniclastica* (4 isolates) were exclusively isolated from layer chicken samples. A comprehensive depiction of the occurrence and frequency of these bacterial isolates in chicken samples is provided in **Fig 1**.

## Phylogenetic analysis of identified bacterial isolates

Phylogenetic analysis utilizing the *16S* rRNA sequencing data of the bacterial isolates reveals distinct clustering patterns (**Fig 3**). Notably, *P. stuartii*, *S. enterica*, and *M. morganii* isolates exhibit a close relationship, forming a tightly grouped cluster. Similarly, *E. coli* (lactose positive), *E. coli* (lactose negative), *K. pneumoniae*, and *E. hormaechei* isolates also form a cohesive cluster, indicating their close relatedness. In contrast, *P. penneri* isolates from the *Enterobacteriaceae* family display a greater genetic distance from the aforementioned bacteria. Additionally, the species *W. chitiniclastica*, belonging to the *Incertae sedis* family, demonstrates a notable divergence from other *Enterobacteriaceae* family members, displaying a distant relationship in the phylogenetic analysis.

## Phenotypic characterization of antibiotic resistant pattern

Upon scrutinizing the antibiogram results, a noteworthy trend emerged. All tested 47 isolates from broiler chickens exhibited resistance to both ampicillin and erythromycin. Among the layer chickens, all tested bacterial isolates demonstrated complete resistance to ampicillin, imipenem, azithromycin, and erythromycin. In the case of broiler chickens, a unique pattern emerged whereby all bacterial isolates displayed sensitivity to ciprofloxacin, norfloxacin, kanamycin, and tetracycline. However, an intermediate state was observed with respect to these drugs, ranging from 19.15% to 97.87% for ciprofloxacin (9 isolates), norfloxacin (8 isolates), kanamycin (4 isolates), and tetracycline (1 isolates), respectively (**Fig 4**).

Meanwhile, bacterial isolates derived from layer chickens showcased a distinctive pattern of sensitivity to cefoxitin and kanamycin but exhibited intermediate effects in 3 isolates (14.29%) and in 1 isolate (4.76%) towards these drugs, respectively. Remarkably, among the bacterial isolates from broiler chickens, gentamicin (21 isolates, 44.68%) and polymyxin B (26 isolates,

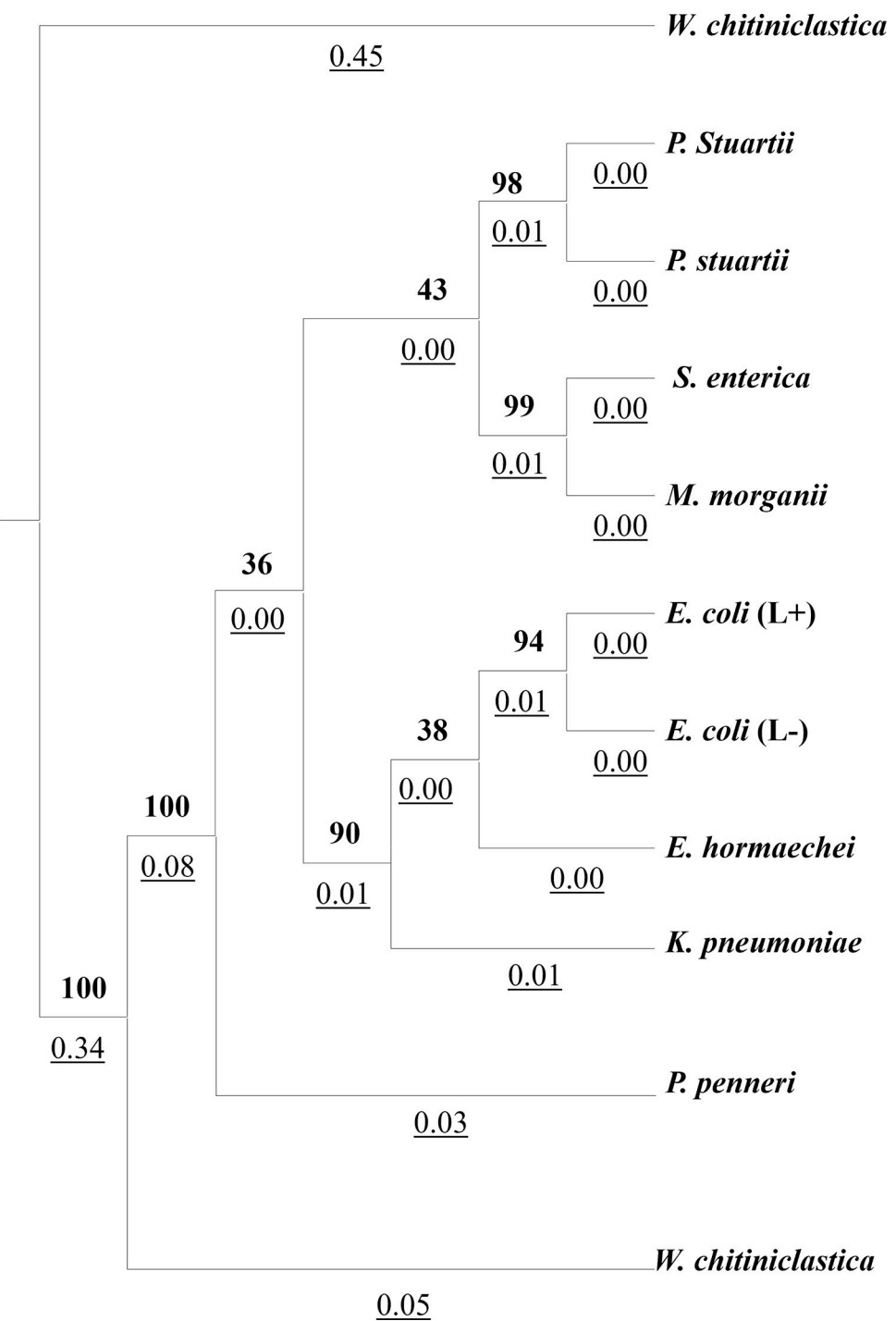

**Fig 3. Phylogenetic tree of identified bacterial isolates (Numeric values with bottom border represent the branch length and numeric values in bold represent the boot strap value).**

52.38%) demonstrated notably higher efficacy compared to other drugs employed in this study. Notably, this study did not uncover any bacterial group exhibiting sensitivity to only one type of drug utilized in this study. The phenotypic antimicrobial traits of bacterial isolates found in both broiler and layer chickens, along with a comparison of antimicrobial

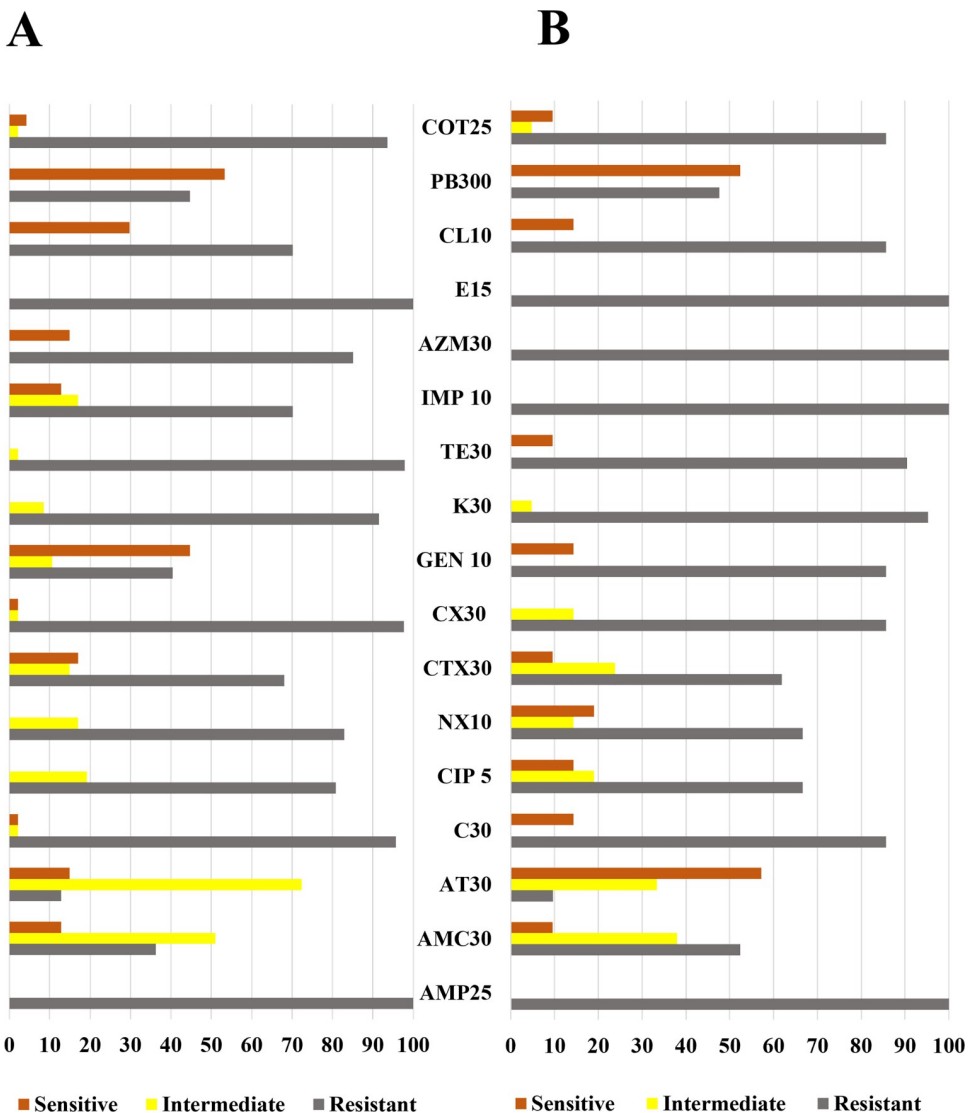

**Fig 4.** Phenotypic identification of antibiotic resistant of total bacterial isolates found in broiler (A) and layer (B) chickens. (Ampicillin (AMP25), Amoxicillin-clavulanic acid (AMC 30), Cefotaxime (CTX 30) Cefoxitin (CX 30) Ciprofloxacin (CIP 5) and Norfloxacin (NX 10), Aztreonam (AT 30), Gentamicin (GEN 10), Kanamycin (K 30), Azithromycin (AZM 30) Erythromycin (E 10), Imipenem (IMP 10), Chloramphenicol (C 30), Trimethoprim-Sulfamethoxazole (COT 25), Tetracycline (TE 30), Colistin (CL 10) and Polymyxin B (PB 300)).

characteristics between the two groups, are elucidated in **Fig 4**. This study reveals a concerning trend: *E. coli*, *K. pneumoniae*, *P. stuartii*, and *P. penneri* isolates from both broiler and layer chicken samples displayed heightened resistance levels compared to those found exclusively in broiler chicken samples. Notably, isolates of *W. chitiniclastica*, newly identified in Bangladeshi poultry chicken samples through this study, exhibited a similarly elevated level of resistance to commonly used antibiotics. We also observed potential polymyxin (colistin and polymyxin-B) resistance among isolates based on their zone of inhibition, measuring less than 12 mm. The implications of resistance to these drugs are grave, as colistin serves as the last resort when other antibiotics fail—a concerning development for the Noakhali region.

Among *E. coli* isolates, AT 30 (12 isolates, 63.18%), GEN 10 (10 isolates, 52. 63%), CL 10 (13 isolates, 68.42%), PB 300 (18 isolates, 92.74%) demonstrated effectiveness in broiler isolates, while AMC 30 (2 isolates, 40%), AT 30 (3 isolates, 60%), PB 300 (5 isolates, 100%) observed high efficacy in layer chicken isolates **S2 Fig**. Conversely, all tested *K. pneumoniae* isolates from broiler chickens exhibited medium level of sensitivity to AT 30 (6 isolates, 81.82%), GEN 10 (4 isolates, 57.14%), PB 300 (4 isolates, 57.14%). Intriguingly, *K. pneumoniae* isolates from layer chickens displayed sensitivity at low to medium level to AMC 30 (1 isolates, 25%), AT 30 (3 isolates, 75%), NX 10 (1 isolates, 25%) drugs only as depicted in **S3 Fig**. Similarly, all *E. hormaechei* isolates from broiler chickens were sensitive to AT 30 (2 isolates, 50%), TE 30 (1 isolates, 25%), PB 300 (3 isolates, 75%) whereas layer chicken isolates showed sensitivity to C 30 (2 isolates, 100%), CIP 5 (2 isolates, 100%), NX 10 (2 isolates, 100%), GEN 10 (2 isolates, 100%), TE 30 (2 isolates, 100%), PB 300 (2 isolates, 100%), COT 25 (2 isolates, 100%) demonstrated in **S4 Fig**. Furthermore, *P. penneri* isolates from broiler chicken demonstrated sensitivity to AMC 30 (2 isolates, 66.67%), AT 30 (2 isolates, 66.67%), CTX 30 (2 isolates, 66.67%), GEN 10 (1 isolates, 33.33%), and from layer chicken isolates demonstrated sensitivity to AT 30 (3 isolates, 100%), CTX 30 (1 isolates, 33.33%) (**S5 Fig**). One *P. stuartii* and one *Aeromonas spp.* isolate from this study found pan-drug resistant and other isolates show high level of resistance to the used antibiotics. Antibiogram result data of *P. stuartii*, *S. enterica*, *M. morganii*, *W. chitiniclastica*, and *Aeromonas spp.* illustrated in **S6**–**S8 Figs**.

## Analyzing MAR phenotype and MAR index in bacterial isolates from broiler and layer chickens

The bacterial species identified in this study exhibited diverse MAR phenotypes, each characterized by distinct MAR indices. Among *E. coli* isolates from broiler chicks, 14 distinct MAR phenotypes were observed, whereas *E. coli* isolates from layer chicks displayed 2 different MAR phenotypes. *K. pneumoniae* isolates from broiler and layer chicks demonstrated 7 and 3 distinct MAR phenotypes, respectively. Similarly, *E. hormaechei* and *P. penneri* isolates from both broiler and layer chick samples exhibited 3 and 2 MAR phenotypes, respectively. Notably, *P. stuartii* and *S. enterica* isolates from broiler chicks displayed 9 and 4 different MAR phenotypes, respectively. Intriguingly, a single *P. stuartii* and *Aeromonas spp.* isolate exhibited resistance to all 17 drugs tested in this study, classifying it as pan-drug resistant. In contrast, *M. morganii*, *W. chitiniclastica*, and *Aeromonas spp.* isolates from layer chicks displayed 2, 3, and 3 different MAR phenotypes, respectively. Particularly concerning is the high level of resistance exhibited by *W. chitiniclastica* isolates, underscoring a worrisome state of antibiotic resistance. Detailed information regarding MAR phenotypes and MAR indices of the isolated bacterial strains can be found in **Table 2**.

## Genotypic characterization of bacterial isolates from broiler and layer chicken samples

Upon evaluating ESBL gene presence and other genes encoding resistance markers in collected isolates, we observed that lactose-positive and lactose-negative *E. coli* carried *bla*TEM, *tet A*, *tet B*, *sul 1*, and *sul 2* genes, with the latter also harboring *bla*SHV (**Fig 5A**–**5C**). These *E. coli* strains displayed resistance to β-lactam combination agents, penicillin, cephalosporins, tetracycline, and folate pathway antagonists. Similarly, *K. pneumoniae* isolates from broiler and layer chickens showed the presence of *bla*TEM, *bla*SHV, *sul 1*, and *tet A* genes (**Fig 5D**). A significant portion of *K. pneumoniae* isolates from both groups exhibited resistance to β-lactam combination agents, penicillin, cephalosporins, tetracycline, and folate pathway antagonists.

**Table 2. MAR phenotypes and MAR index profile of isolated bacterial isolates from broiler and layer chicken samples.**

| Antibiotic resistant profile | No. of Isolates | MAR index |
|---|---|---|
| ***E. coli* isolate from broiler chickens** | | |
| AMP 25, AMC 30, AT 30, C 30, CIP 5, NX 10, CTX 30, CX30, GEN 10, K 30, TE 30, E 15, CL 10, PB300, COT 25 | 1 | 0.82 |
| AMP 25, C 30, CIP 5, NX 10, CTX 30, CX30, GEN 10, K 30, TE 30, IMP10, AZM 30, E 15, CL 10, COT 25 | 1 | 0.82 |
| AMP 25, AMC 30, C 30, CIP 5, NX 10, CTX 30, CX30, K 30, TE 30, IMP10, AZM 30, E 15, COT 25 | 1 | 0.76 |
| AMP 25, AT 30, C 30, CIP 5, NX 10, CTX 30, CX30, K 30, TE 30, AZM 30, E 15, COT 25 | 1 | 0.71 |
| AMP 25, C 30, CIP 5, NX 10, CX30, GEN 10, K 30, TE 30, AZM 30, E 15, CL 10, COT 25 | 2 | 0.71 |
| AMP 25, C 30, CIP 5, NX 10, CTX 30, CX30, K 30, TE 30, IMP 10, E 15, CL 10, COT 25 | 1 | 0.71 |
| AMP 25, C 30, CIP 5, NX 10, CX30, GEN 10, K 30, TE 30, E 15, CL 10, COT 25 | 2 | 0.65 |
| AMP 25, C 30, CIP 5, NX 10, CTX 30, CX30, K 30, TE 30, AZM 30, E 15, COT 25 | 1 | 0.65 |
| AMP 25, C 30, CIP 5, NX 10, CTX 30, K 30, TE 30, IMP10, AZM 30, E 15, COT 25 | 2 | 0.65 |
| AMP 25, C30, CIP 5, NX 10, CX 30, CTX30, K 30, TE 30, E 15, COT 25 | 1 | 0.59 |
| AMP 25, CIP 5, NX 10, CX 30, TE 30, IMP10, AZM 30, E 15, COT 25 | 1 | 0.53 |
| AMP 25, C30, CIP 5, NX 10, CX 30, K 30, TE 30, AZM 30, E 15. | 1 | 0.53 |
| AMP 25, AT 30, C30, NX 10, CTX 30, CX 30, K 30, E 15 | 1 | 0.47 |
| AMP 25, AT 30, C30, CIP5, NX 10, K 30, TE 30, AZM 30, E 15, | 1 | 0.47 |
| ***E. coli* isolates from layer chickens** | | |
| AMP 25, C 30, CIP 5, NX 10, CTX 30, CX30, GEN 10, K 30, TE 30, IMP 10, AZM 30, E 15, CL 10, COT 25 | 3 | 0.82 |
| AMP 25, C 30, CIP 5, NX 10, CTX 30, CX30, GEN 10, K 30, TE 30, IMP 10, AZM 30, E 15, CL 10, COT 25 | 1 | 0.76 |
| ***K. pneumoniae* isolates from broiler chickens** | | |
| AMP 25, AMC 30, C 30, CIP 5, CTX 30, CX30, GEN 10, K 30, TE 30, IMP 10, AZM 30, E 15, CL 10, PB300, COT 25 | 1 | 0.88 |
| AMP 25, AMC 30, C 30, CIP 5, NX 10, CTX 30, CX30, TE 30, IMP10, AZM 30, E 15, CL 10, PB 300, COT 25 | 1 | 0.82 |
| AMP 25, C 30, CIP 5, NX 10, CTX 30, CX30, GEN 10, K 30, TE 30, IMP 10, AZM 30, E 15, CL 10, COT 25 | 1 | 0.82 |
| AMP 25, AMC 30, C 30, CIP 5, CTX 30, CX30, TE 30, IMP 10, AZM 30, E 15, CL 10, PB 300, COT 25 | 1 | 0.76 |
| AMP 25, C 30, CIP 5, NX 10, CX30, K 30, TE 30, IMP 10, AZM 30, E 15, CL 10, COT 25 | 1 | 0.71 |
| AMP 25, C 30, CIP 5, NX 10, CX30, K 30, TE 30, AZM 30, E 15, CL 10, COT 25 | 1 | 0.65 |
| AMP 25, C 30, CIP 5, NX 10, CX30, GEN 10, K 30, TE 30, AZM 30, E 15, COT 25 | 1 | 0.65 |
| ***K. pneumoniae* isolates from layer chickens** | | |
| AMP 25, AMC 30, C 30, CIP 5, NX 10, CTX 30, CX30, GEN 10, K 30, TE 30, IMP 10, AZM 30, E 15, CL 10, PB300, COT 25 | 1 | 0.94 |
| AMP 25, AMC 30, C 30, CTX 30, GEN 10, K 30, TE 30, IMP 10, AZM 30, E 15, CL10, PB 300, COT 25 | 2 | 0.76 |
| AMP 25, AMC 30, CX 30, GEN 10, K 30, TE 30, IMP 10, AZM 30, E 15, CL10, PB 300 | 1 | 0.64 |
| ***E. hormaechei* isolates from broiler chickens** | | |
| AMP 25, AMC 30, C 30, CIP 5, NX 10, CTX 30, CX30, GEN 10, K 30, TE 30, IMP 10, AZM 30, E 15, CL 10, PB300, COT 25 | 1 | 0.94 |
| AMP 25, C 30, CIP 5, NX 10, CTX 30, CX 30, GEN 10, K 30, TE 30, IMP 10, AZM 30, E 15, CL10, COT 25 | 2 | 0.82 |
| AMP 25, C 30, CIP 5, NX 10, CTX 30, CX 30, GEN 10, TE 30, IMP 10, AZM 30, E 15, CL10, COT 25 | 1 | 0.76 |
| ***E. hormaechei* isolates from layer chickens** | | |

(*Continued*)

**Table 2.** (Continued)

| Antibiotic resistant profile | No. of Isolates | MAR index |
|---|---|---|
| AMP 25, AMC 30, CTX 30, CX30, IMP 10, AZM 30, E 15, CL 10 | 1 | 0.47 |
| AMP 25, AMC 30, CTX 30, CX30, E 15, CL 10 | 1 | 0.35 |
| *P. penneri* isolates from broiler chickens | | |
| AMP 25, C 30, CIP 5, NX 10, CTX 30, CX30, K 30, TE 30, IMP 10, AZM 30, E 15, CL 10, PB300, COT 25 | 1 | 0.82 |
| AMP 25, AMC 30, C 30, CIP 5, NX 10, CX30, K 30, TE 30, IMP 10, AZM 30, E 15, CL 10, PB300, COT 25 | 1 | 0.82 |
| AMP 25, CIP 5, NX 10, CX30, GEN 10, K 30, TE 30, IMP 10, AZM 30, E 15, CL 10, PB300, COT 25 | 1 | 0.76 |
| *P. penneri* isolates from layer chickens | | |
| AMP 25, AMC 30, C 30, CIP 5, NX 10, CX30, GEN 10, K 30, TE 30, IMP 10, AZM 30, E 15, CL 10, PB300, COT 25 | 1 | 0.88 |
| AMP 25, C 30, CIP 5, NX 10, CX30, GEN 10, K 30, TE 30, IMP 10, AZM 30, E 15, CL 10, PB300, COT 25 | 2 | 0.82 |
| *P. stuartii* isolates from broiler chickens | | |
| AMP 25, AMC30, AT 30, C 30, CIP 5, NX 10, CTX 30, CX30, K 30, TE 30, IMP 10, AZM 30, E 15, CL 10, PB300, COT 25 | 1 | 1.0 |
| AMP 25, AMC 30, C 30, NX 10, CTX 30, CX30, GEN 10, K 30, TE 30, IMP 10, AZM 30, E 15, CL 10, PB300, COT 25 | 1 | 0.88 |
| AMP 25, AMC 30, C 30, CIP 5, CTX 30, CX30, GEN 10, K 30, TE 30, IMP 10, AZM 30, E 15, CL 10, PB300, COT 25 | 1 | 0.88 |
| AMP 25, AMC 30, C 30, CIP 5, NX 10, CTX 30, CX30, K 30, TE 30, IMP 10, AZM 30, E 15, CL 10, PB300, COT 25 | 1 | 0.88 |
| AMP 25, AMC 30, AT 30, C 30, CTX, CX30, K 30, TE 30, IMP 10, AZM 30, E 15, CL 10, PB300, COT 25 | 1 | 0,82 |
| AMP 25, AMC 30, C 30, CTX, CX30, GEN 10, K 30, TE 30, IMP 10, AZM 30, E 15, CL 10, PB300, COT 25 | 1 | 0.82 |
| AMP 25, AMC 30, C 30, CTX, CX30, K 30, TE 30, IMP 10, AZM 30, E 15, CL 10, PB300, COT 25 | 2 | 0.76 |
| AMP 25, C 30, NX 10, CX 30, K 30, TE 30, IMP 10, AZM 30, E 15, CL 10, PB300, COT 25 | 1 | 0.71 |
| AMP 25, AMC 30, C 30, CX 30, K 30, TE 30, AZM 30, E 15, CL 10, PB300, COT 25 | 1 | 0.64 |
| *S. enterica* isolates from broiler chickens | | |
| AMP 25, AMC 30, C 30, NX 10, CTX 30, CX30, GEN 10, K 30, TE 30, IMP 10, AZM 30, E 15, CL 10, PB300, COT 25 | 1 | 0.88 |
| AMP 25, C 30, CIP 5, NX 10, CTX 30, CX30, GEN 10, K 30, TE 30, IMP 10, AZM 30, E 15, CL 10, PB300, COT 25 | 1 | 0.88 |
| AMP 25, C 30, CIP 5, NX 10, CTX 30, CX30, GEN 10, K 30, TE 30, AZM 30, E 15, CL 10, PB300, COT 25 | 1 | 0.82 |
| AMP 25, C 30, CIP 5, NX 10, CTX 30, CX30, K 30, TE 30, IMP10, AZM 30, E 15, CL 10, COT 25 | 1 | 0.76 |
| *M. morganii* isolates from layer chickens | | |
| AMP 25, C 30, CIP 5, NX 10, CTX 30, CX30, GEN 10, K 30, TE 30, IMP 10, AZM 30, E 15, CL 10, COT 25 | 1 | 0.82 |
| AMP 25, C 30, CX30, K 30, TE 30, IMP 10, AZM 30, E 15, CL 10, PB 300, COT 25 | 1 | 0.65 |
| *W. chitiniclastica* isolates from layer chickens | | |
| AMP 25, AMC 30, AT 30, C 30, CIP 5, NX 10, CX30, GEN 10, K 30, TE 30, IMP 10, AZM 30, E 15, CL 10, COT 25 | 1 | 0.88 |
| AMP 25, AMC 30, C 30, CIP 5, NX 10, CX30, GEN 10, K 30, TE 30, IMP 10, AZM 30, E 15, CL 10, COT 25 | 1 | 0.82 |

(*Continued*)

**Table 2.** (Continued)

| Antibiotic resistant profile | No. of Isolates | MAR index |
|---|---|---|
| AMP 25, AMC 30, C 30, CIP 5, NX 10, CX30, GEN 10, K 30, TE 30, IMP 10, AZM 30, E 15, COT 25 | 1 | 0.76 |
| *Aeromonas spp.* isolates from layer chickens | | |
| AMP 25, AMC30, AT 30, C 30, CIP 5, NX 10, CTX 30, CX30, K 30, TE 30, IMP 10, AZM 30, E 15, CL 10, PB300, COT 25 | 1 | 1.0 |
| AMP 25, C 30, CTX 30, CX30, GEN 10, K 30, TE 30, IMP 10, AZM 30, E 15, CL 10, PB 300, COT 25 | 1 | 0.76 |
| AMP 25, AMC 30, K 30, IMP 10, AZM 30, E 15 | 1 | 0.35 |

Among the isolates, *E. hormaechei*, *P. penneri*, *P. stuartii*, *W. chitiniclastica*, and *M. morganii* carried *tet A*, *sul 1*, *bla*SHV, and *bla*TEM genes (**Fig 5E–5G, 5I, 5J**) aligning with their phenotypic resistance.

Lastly, *S. enterica* isolates carried *sul 1*, *sul 2*, *bla*TEM genes (**Fig 5H**) and most of the *S. enterica* isolates shown resistance to β-lactam combination agents, penicillin, cephalosporins, tetracycline, and folate pathway antagonists. As those isolates showed resistance to tetracycline without the presence of both *tet A* and *tet B* genes indicated that the presence of other genes encoding tetracycline resistance, which were not investigated in this study, in *S. enterica* isolates.

Along with those eight different types of antibiotic-resistant genes, isolates of lactose-negative *E. coli*, *K. pneumoniae*, *E. hormaechei*, *P. stuartii*, and *M. morganii* were also found to carry the *mcr-1* variant of the colistin-resistant gene (**Fig 6**).

Co-relation between phenotypic and genotypic antibiotic resistant showed that identified resistant genes were strongly associated with the phenotypic effect to show resistant against different drugs illustrated in **Fig 7**. Standard zone of inhibition (ZOI) value to interpret the antibiogram result given in **S13 Table**. Moreover, *E. coli* (lactose positive) *S. enterica* and *W. chitiniclastica* showed resistance to colistin but didn't carry *mcr-1* gene (**Fig 7**). This may be due to presence of other variant of *mcr* gene.

## Discussion

Poultry production is a highly significant food industry, accounting for approximately 90 billion tons of chicken meat produced annually worldwide [52]. To promote poultry growth, many countries rely on a diverse range of antimicrobials [53–55], some of which are considered essential in human medicine [56, 57]. However, the unrestricted use of these critical antimicrobials in animal production contributes to the rise of antimicrobial resistance (AR) in both commensal and pathogenic microbes. Besides, AR characteristics of pathogenic bacteria are continuously changing every year. This alarming situation can lead to treatment failures, financial losses, and the potential transmission of resistant genes to humans. Furthermore, concerns about human health arise due to the presence of antibiotic residues in meat, eggs, and other animal products [58, 59]. To address this issue, we conducted this study, and our focus was on evaluating multi-drug resistant (MDR) *Enterobacteriaceae* and Gram-negative non-*Enterobacteriaceae* family bacteria from poultry chickens from Noakhali region of Bangladesh, known to be causative agents of different diseases in both humans and animals.

The prevalent bacteria included *E. coli* (Lactose positive and lactose negative), *K. pneumoniae*, *E. hormaechei*, and *P. penneri*. Additionally, *Providencia spp.* and *S. enterica* were found exclusively in broiler chickens, while *M. morganii*, *W. chitiniclastica*, and *Aeromonas spp.* were

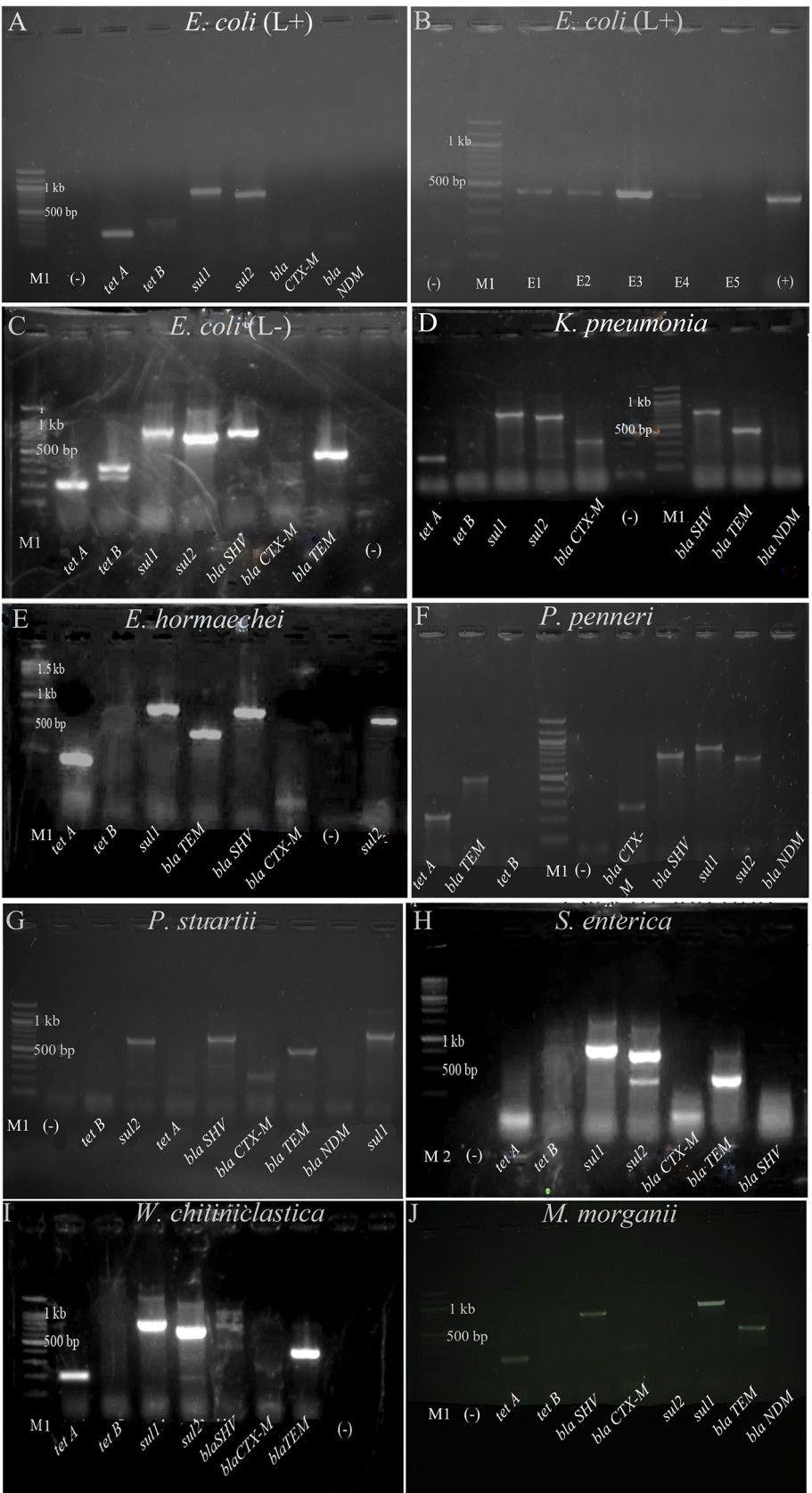

**Fig 5.** Detection of antibiotic resistant genes from A) *E. coli* (lactose positive) B) *bla*TEM gene for lactose positive *E. coli* C) *E. coli* (lactose negative) D) *K. pneumoniae*, E) *E. hormaechei*, F) *P. penneri*, G) *P. stuartii*, H) *S. enterica*, I) *W. chitiniclastica*, and J) *M. morganii*, (M1 = 100 bp DNA ladder, M2 = 250 bp ladder for **E and H**, E = *E. coli*, (-) = negative control, (L+) = Lactose positive, (L-) = Lactose negative, *tet A* = 201 bp, *tet B* = 359 bp, *sul1* = 822 bp, *sul2* = 625 bp, *bla*TEM = 445 bp, *bla*CTX-*M* = 593 bp, *bla*SHV = 747 bp, *bla*NDM = 621 bp. We have found *bla*CTX-M in *K. pneumoniae*, *P. stuartii*, *M. morganii*, and *P. penneri*.at 350 bp approximately).

detected in stool and rectal swab samples from layer chickens. These isolates have been previously recognized as pathogens that can cause diseases in both humans and animals and align with previous research conducted in Bangladesh and globally [60–68]. To the best of our knowledge, Lactose negative *E. coli*, *M. morganii*, *E. hormaechei and W. chitiniclastica* were identified for the first time in Bangladeshi poultry chicken Gut sample. MDR Lactose negative *E. coli* isolates found from human gut can cause different types of intestinal and non-intestinal infections as reported by one study conducted in Bangladesh [69].

After identifying the isolates, we conducted a phenotypic analysis of antibiotic resistance in the bacterial samples obtained from both broiler and layer chickens. Our findings revealed, in broiler chickens all *E. coli* isolates demonstrated resistance to ampicillin, norfloxacin, and erythromycin. Additionally, *E. coli* isolates exhibited intermediate effects towards ciprofloxacin (1 isolate, 94.74%), kanamycin (1 isolate, 94.74%), and tetracycline (1 isolate, 94.74%) with 0% sensitivity which means those isolates can become resistant in near future. However, we also identified colistin-resistant *E. coli* (mostly in lactose negative) isolates, which were primarily assessed using the agar disk diffusion method, showing either no zone of inhibition or a zone of inhibition below 12 mm according to Uwizeyimana JD. *et al* (2020) and Fadare FT *et. al.* (2021) [47, 48]. All *E. coli* isolates from the layer chicken samples exhibited resistance to

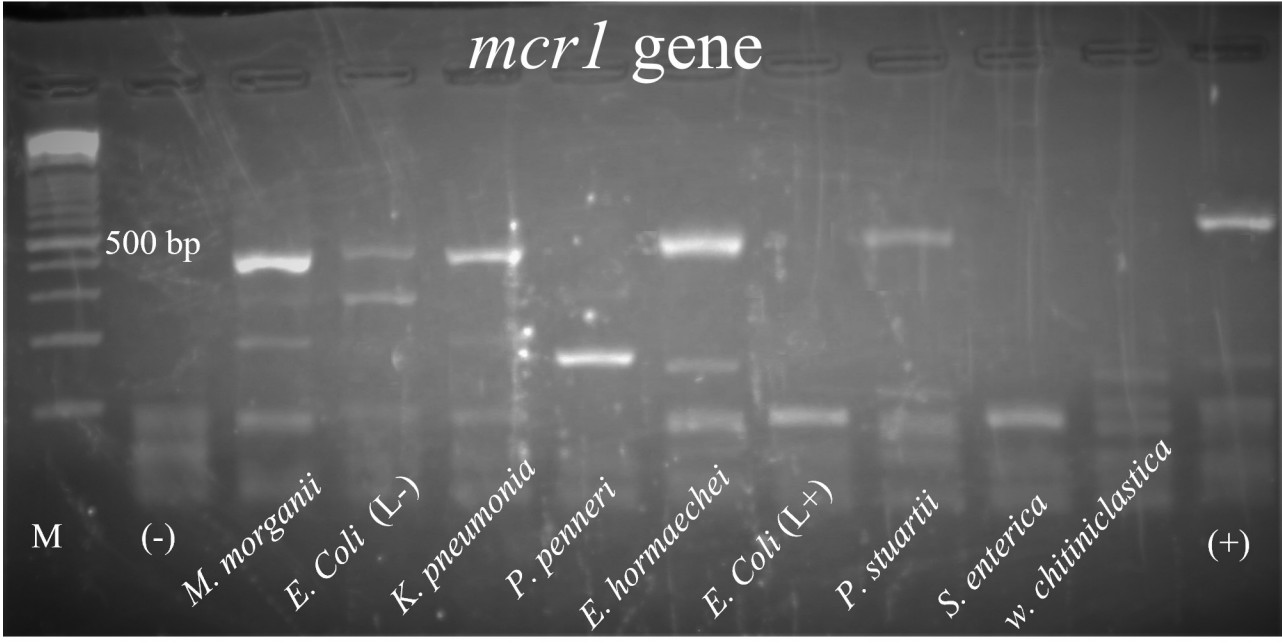

**Fig 6. Detection of *mcr-1* gene from *E. coli* (lactose positive), *E. coli* (lactose negative), *K. pneumoniae*, *E. hormaechei*, *P. penneri*, *P. stuartii*, *S. enterica*, *W. chitiniclastica*, and *M. morganii* isolates.** (M = 100 bp DNA ladder, (-) = negative control, (L+) = Lactose positive, (L-) = Lactose negative, (+) = Positive control, band size for *mcr-1* gene was 309 bp).

| Organism | Antibiotic-resistant genes | Antibiotics | | | | | | | | |
|---|---|---|---|---|---|---|---|---|---|---|
| | ZOI=0 | AMP 30 | AMC 30 | CX 30 | CTX 30 | AT 30 | IMP 10 | TE 30 | COT 25 | CL 10 |
| *E. coli* (L+) | *bla*TEM, *tet A, tet B, sul1, sul 2* | R, ZOI=0 | S, ZOI=19 | R, ZOI=0 | S, ZOI=28 | S, ZOI=29 | S, ZOI=23 | R, ZOI=0 | R, ZOI=0 | R, ZOI<12 |
| *E. coli* (L-) | *bla*TEM, *tet A, tet B, sul 1, and sul 2, mcr-1* | R ZOI=0 | I, ZOI=17 | R, ZOI=0 | S, ZOI=26 | S, ZOI=30 | R, ZOI=14 | R, ZOI=0 | R, ZOI=0 | R, ZOI=0 |
| *K. pneumoniae* | *bla*SHV, *bla*TEM, *tet A, Sul 1, mcr-1* | R, ZOI=0 | I ZOI=15 | R, ZOI=0 | R, ZOI=16 | R, ZOI=12 | R, ZOI=16 | R, ZOI=0 | R, ZOI=0 | R, ZOI=0 |
| *P. stuartii* | *bla*SHV, *bla*TEM, *tet A, sul1, sul 2, mcr-1* | R, ZOI=0 | R, ZOI=9 | R, ZOI=18 | R, ZOI=0 | R, ZOI=12 | R, ZOI=0 | R, ZOI=0 | R, ZOI=0 | R, ZOI=0 |
| *E. hormaechei* | *tet A, sul 1, bla*SHV, *and bla*TEM | R ZOI=0 | R, ZOI=0 | R, ZOI=0 | S, ZOI=28 | S, ZOI=29 | R, ZOI=0 | R, ZOI=0 | R, ZOI=0 | R, ZOI=0 |
| *P. penneri* | *bla*SHV, *bla*TEM, *tet A, sul 1, sul 2* | R, ZOI=0 | I, ZOI=15 | R, ZOI=0 | S, ZOI=30 | S, ZOI=30 | R, ZOI=0 | R, ZOI=0 | R, ZOI=0 | R, ZOI<12 |
| *M. morganii* | *bla*SHV, *bla*TEM, *tet A, sul 1, mcr-1* | R, ZOI=0 | R, ZOI=8 | R, ZOI=0 | S, ZOI=27 | S, ZOI=28 | R, ZOI=0 | R, ZOI=0 | R, ZOI=0 | R, ZOI=0 |
| *W. chitiniclastica* | *tet A, sul 1, bla*SHV, *and bla*TEM, | R, ZOI=0 | R, ZOI=13 | R, ZOI=0 | S, ZOI=28 | R, ZOI=13 | R, ZOI=0 | R, ZOI=0 | R, ZOI | R, ZOI<12 |
| *S. enterica* | *sul 1, sul 2, bla*TEM | R, ZOI=0 | R, ZOI=9 | R, ZOI=0 | I, ZOI=23 | S, ZOI=27 | R, ZOI=8 | S, ZOI=19 | R, ZOI=0 | R, ZOI<12 |

**Fig 7. Phenotypic and genotypic co-relation of bacterial isolates.** (Ampicillin (AMP25), Amoxicillin-clavulanic acid (AMC 30), Cefotaxime (CTX 30) Cefoxitin (CX 30), Aztreonam (AT 30), Imipenem (IMP 10), Tetracycline (TE 30), Trimethoprim-Sulfamethoxazole (COT 25), Colistin (CL 10), ZOI<10 = Zone of inhibition is less than 10 mm).

multiple antibiotics, including ampicillin, chloramphenicol, ciprofloxacin, norfloxacin, cefotaxime, cefoxitin, gentamicin, kanamycin, tetracycline, imipenem, azithromycin, erythromycin, and Co-Trimoxazole. In comparison to broiler, (4 isolates, 80%) of *E. coli* isolates from layer chickens were resistant to colistin, and some of these isolates (mostly lactose negative *E. coli*) showed no zone of inhibition. The results from our study indicating a higher percentage of *E. coli* isolates showing resistance to colistin in both broiler (6 isolates, 38%) and layer (4 isolates, 80%) chickens compared to the previous study conducted in Bangladesh are indeed concerning [70–72].

All the *K. pneumoniae* isolates of this study from broiler chickens exhibited resistance to multiple antibiotics, including ampicillin, chloramphenicol, ciprofloxacin, cefoxitin, tetracycline, azithromycin, erythromycin, and Co-Trimoxazole, classifying them as multi-drug resistant (MDR). Among the tested antibiotics, Aztreonam (6 isolates, 86%), gentamicin (4 isolates,

57%), and Polymyxin B (4 isolates, 57%) were found to be the most effective against *K. pneumoniae* isolates in broiler chickens. In contrast, *K. pneumoniae* isolates from layer chicken samples demonstrated resistance to ampicillin, gentamicin, kanamycin, tetracycline, imipenem, azithromycin, erythromycin, colistin, and polymyxin B. Additionally, 1 isolate (25%) of *K. pneumoniae* showed intermediate effects, and 3 isolates (75%) were resistant to Co-Trimoxazole, suggesting that Co-Trimoxazole resistance might become more prevalent in the near future. These findings indicate a substantial increase in antibiotic-resistant *K. pneumoniae* compared to previously reported studies on poultry chickens and antibiotic-resistant bacteria [73–76].

In our study, *E. hormaechei* isolates from both broiler and layer samples exhibited resistance to seven and four classes of antibiotics, respectively, which was previously unreported in Bangladesh. Interestingly, *E. hormaechei* from layer samples displayed higher sensitivity to chloramphenicol, ciprofloxacin, norfloxacin, gentamicin, tetracycline, and polymyxin B compared to broiler samples. On the other hand, all *P. penneri* isolates from both broiler and layer samples demonstrated resistance to multiple antibiotics, including ampicillin, ciprofloxacin, norfloxacin, cefoxitin, kanamycin, tetracycline, imipenem, azithromycin, erythromycin, colistin, polymyxin B, and Co-Trimoxazole. In broiler samples, 2 isolates (67%) of *P. penneri* were sensitive to amoxicillin-clavulanic acid and aztreonam, whereas in layer samples, no isolates were sensitive to amoxicillin-clavulanic acid, and 2 isolates (67%) showed intermediate effects, while all isolates were resistant to aztreonam. Additionally, 2 (66.67%) of *P. penneri* isolates from broiler samples showed sensitivity to cefotaxime, while in layer samples, 1 (33%) of isolates exhibited sensitivity to this antibiotic.

Several studies on MDR *S. enterica* and *P. stuartii* have reported resistance patterns in broiler samples from Dhaka, Gazipur, Sherpur, Mymensingh, and Chattogram. In these regions, *S. enterica* isolates exhibited resistance to various antibiotics, such as Penicillin-g (90–100%), ampicillin (82.85–100%), amoxicillin (90–98%), cephalexin (70%), streptomycin (77.14%), tetracycline (93–97.14%), chloramphenicol (94.28%), cotrimoxazole (80%), nitrofurantoin (50–78%), sulfamethoxazole (60%), gentamicin (40–46%), erythromycin (80%), nalidixic acid (40–66.6%), kanamycin (40–80%), doxycycline (66.66%), ciprofloxacin (20–40%), and imipenem (83.33%) [63, 77–82]. In comparison, the *S. enterica* isolates obtained in our study exhibited higher levels of resistance to the antibiotics tested compared to those in previous studies. All *P. stuartii* isolates displayed resistance to ten antibiotics from nine different classes out of the seventeen antibiotics tested across twelve classes. In our study, 7 (70%), 1 (10%), and 5 isolates (50%) of *P. stuartii* isolates exhibited resistance to aztreonam, cefotaxime, and gentamicin, respectively. Regarding *S. enterica* isolates, all of them demonstrated resistance to nine antibiotics from twelve classes tested, and 2 (50%) of these isolates were also resistant to azithromycin. However, all *S. enterica* isolates in this study were sensitive to aztreonam.

*M. morganii*, *Aeromonas spp*., and *W. chitiniclastica* isolates were exclusively detected in layer chicken samples, with *M. morganii* and *W. chitiniclastica* being isolated for the first time in Bangladesh. Previous studies reported concerning rates of multi-drug resistance (MDR) in *M. morganii*, with 54% in poultry chicken meat from Tennessee [62] and 52% in poultry chickens in Nigeria [83]. In our study, we found 2 isolates (100%) of *M. morganii* to be resistant to ampicillin, cefoxitin, kanamycin, tetracycline, imipenem, azithromycin, erythromycin, colistin, and Co-Trimoxazole, posing an alarming situation for the Noakhali region of Bangladesh.

In our study, *W. chitiniclastica* strains isolated from layer chicken samples displayed resistance to all antibiotics used, except aztreonam and polymyxin B. Moreover, 2 isolates (66.7%) of these showed intermediate effects towards aztreonam. *W. chitiniclastica* has been recognized as an emerging zoonotic pathogen by the United States Centers for Disease Control and

Prevention (CDC) [84]. Kopf et al. suggested the use of trimethoprim/sulfamethoxazole, levofloxacin, and cephalosporins (e.g., ceftazidime) antibiotics for *W. chitiniclastica* infections [65]. However, in our study, we found that all isolates of *W. chitiniclastica* had already developed resistance to these drugs, indicating that these antibiotics may not be effective for treating *W. chitiniclastica* infections.

Alam *et al.* (2010) reported on *Aeromonas spp.* isolated from poultry chicken samples, showing resistance to erythromycin, (16%), norfloxacin (16%), nalidixic acid (15%), tetracycline (15%), ampicillin (10%), and gentamicin (20%), while being sensitive to streptomycin (30%), chloramphenicol (65%), ciprofloxacin (18%), norfloxacin (83%), nalidixic acid (85%), tetracycline (20%), rifampicin (25%), and gentamicin (80%) [85]. Moreover, one other study from Igbinosa *et. al.* (2014) found that all *Aeromonas spp.* from poultry chicken fecal samples were sensitive to ciprofloxacin, gentamicin, and tetracycline [86]. Conversely, in a study on poultry feed used in Bangladesh's poultry farms, *Aeromona*s spp. isolates displayed resistance to rifampin (30%), gentamicin (40%), erythromycin (100%), ceftriaxone (10%), kanamycin (10%), novobiocin (40%), nalidixic acid (10%), amoxicillin (30%), and ciprofloxacin (45%) [87]. In comparison, the isolates from our study exhibited a higher rate of resistance to antibiotics. We observed that all *Aeromonas spp.* isolates showed resistance to ampicillin, kanamycin, imipenem, azithromycin, and erythromycin. Additionally, *Aeromonas spp.* showed resistance to amoxicillin-clavulanic acid (2 isolates, 66.67%), aztreonam (1 isolate, 33.33%), chloramphenicol (2 isolates, 66.67%), ciprofloxacin (2 isolates, 66.67%), norfloxacin (1 isolate, 33.33%), cefotaxime (2 isolates, 66.67%), gentamicin (2 isolates, 66.67%), tetracycline (2 isolates, 66.67%), colistin (2 isolates, 66.67%), polymyxin B (2 isolates, 66.67%), and Co-Trimoxazole (2 isolates, 66.67%).

Chicken farms serve as reservoirs of multi-drug resistant genes worldwide, including Bangladesh [14, 17, 18, 28, 29, 88–91]. MDR genes, such as ESBL genes, AmpC producing genes, carbapenem-resistant genes, colistin-resistant genes, tetracycline, and sulfonamide-resistant genes, have been reported in poultry and poultry products in different regions of Bangladesh [8, 60, 61, 77, 79, 81, 92]. In our study, we screened for ESBL and carbapenem-resistant genes (*bla*CTX-M, *bla*SHV, *bla*TEM, *bla*NDM), tetracycline-resistant genes (*tetA*, *tetB*), and sulfonamide-resistant genes (*sul1*, *sul2*) in the isolates from poultry chicken samples to understand their genotypic characteristics. The identified genes included *bla*SHV, *bla*TEM, *tetA*, and *sul2* in all isolates, while *E. coli*, *K. pneumoniae*, and *P. penneri* isolates contained *tetB* genes. However, the *bla*CTX-M gene was not found in any isolates, but an unwanted band approximately 350 bp was observed, with the actual band size for *bla*CTX-M being 593 bp [93]. The presence of ESBL, tetracycline, and sulfonamide resistant genes in bacterial isolates led to resistance against ampicillin, amoxicillin-clavulanic acid, aztreonam, cefotaxime, cefoxitin, tetracycline, and Co-Trimoxazole. Notably, ESBL genes (*bla*CTX-M, *bla*SHV, *bla*TEM) found on sizable conjugative plasmids are responsible for resistance to other classes of antibiotics like fluoroquinolones, aminoglycosides, and trimethoprim-sulfamethoxazole [94–96]. This highlights the importance of monitoring and controlling the spread of antibiotic resistance in poultry settings to mitigate its impact on public health.

## Conclusion

Poultry and poultry products, particularly in the Noakhali region of Bangladesh, have become reservoirs of multi-drug resistant (MDR) isolates, joining clinical sources in this role. Studies in Bangladesh have focused on MDR isolates such as *E. coli*, *K. pneumoniae*, *Salmonella spp.*, *Campylobacter spp.*, *and Citrobacter spp.* from poultry chickens and farms. Our study identified *Enterobacteriaceae* isolates, including *M. morganii*, showing higher resistance to common

drugs and acting as pathogenic agents for humans and animals, contributing to the spread of antibiotic resistance genes in the region. Additionally, this study assessed antibiotic resistance in *Aeromonas spp.*, *and W. chitiniclastica* from the non-*Enterobacteriaceae* family.

However, the study has limitations. Firstly, the sample collection was limited to four small-scale farms in Sadar Upazila of Noakhali, potentially limiting the comprehensive understanding of MDR bacteria in poultry chickens. Fourth-generation cephalosporins were also not part of the study. Only one variant of the *mcr* gene was evaluated, and other genes responsible for carbapenem, aminoglycoside, and macrolide resistance were not included. Nonetheless, the study highlights the rapid development of antibiotic resistance in both *Enterobacteriaceae* and non-*Enterobacteriaceae* bacterial strains. The high antibiotic use in the poultry industry remains a major concern, posing economic and health risks to both humans and animals.

## Supporting information

**S1 Fig. Colony comparison of *E. coli* isolates from broiler chicken sample.** ((+) = Lactose Positive, (-) = Lactose negative Mac = MacConkey agar).
(TIFF)

**S2 Fig. Phenotypic identification of antibiotic resistant of *E. coli* isolates found from broiler and layer chickens.** (Ampicillin (AMP25), Amoxicillin-clavulanic acid (AMC 30), Cefotaxime (CTX 30) Cefoxitin (CX 30) Ciprofloxacin (CIP 5) and Norfloxacin (NX 10), Aztreonam (AT 30), Gentamicin (GEN 10), Kanamycin (K 30), Azithromycin (AZM 30) Erythromycin (E 10), Imipenem (IMP 10), Chloramphenicol (C 30), Trimethoprim-Sulfamethoxazole (COT 25), Tetracycline (TE 30), Colistin (CL 10) and Polymyxin B (PB 300)).
(TIF)

**S3 Fig. Phenotypic identification of antibiotic resistant of *K. pneumoniae* isolates found from broiler and layer chickens.** (Ampicillin (AMP25), Amoxicillin-clavulanic acid (AMC 30), Cefotaxime (CTX 30) Cefoxitin (CX 30) Ciprofloxacin (CIP 5) and Norfloxacin (NX 10), Aztreonam (AT 30), Gentamicin (GEN 10), Kanamycin (K 30), Azithromycin (AZM 30) Erythromycin (E 10), Imipenem (IMP 10), Chloramphenicol (C 30), Trimethoprim-Sulfamethoxazole (COT 25), Tetracycline (TE 30), Colistin (CL 10) and Polymyxin B (PB 300)).
(TIF)

**S4 Fig. Phenotypic identification of antibiotic resistant of *E. hormaechei* isolates found from broiler and layer chickens.** (Ampicillin (AMP25), Amoxicillin-clavulanic acid (AMC 30), Cefotaxime (CTX 30) Cefoxitin (CX 30) Ciprofloxacin (CIP 5) and Norfloxacin (NX 10), Aztreonam (AT 30), Gentamicin (GEN 10), Kanamycin (K 30), Azithromycin (AZM 30) Erythromycin (E 10), Imipenem (IMP 10), Chloramphenicol (C 30), Trimethoprim-Sulfamethoxazole (COT 25), Tetracycline (TE 30), Colistin (CL 10) and Polymyxin B (PB 300)).
(TIF)

**S5 Fig. Phenotypic identification of antibiotic resistant of *P. penneri* isolates found from broiler and layer chickens.** (Ampicillin (AMP25), Amoxicillin-clavulanic acid (AMC 30), Cefotaxime (CTX 30) Cefoxitin (CX 30) Ciprofloxacin (CIP 5) and Norfloxacin (NX 10), Aztreonam (AT 30), Gentamicin (GEN 10), Kanamycin (K 30), Azithromycin (AZM 30) Erythromycin (E 10), Imipenem (IMP 10), Chloramphenicol (C 30), Trimethoprim-Sulfamethoxazole (COT 25), Tetracycline (TE 30), Colistin (CL 10) and Polymyxin B (PB 300)).
(TIF)

**S6 Fig. Phenotypic identification of antibiotic resistant of *P. stuartii and S. enterica* isolates found from broiler chickens.** (Ampicillin (AMP25), Amoxicillin-clavulanic acid (AMC 30),

Cefotaxime (CTX 30) Cefoxitin (CX 30) Ciprofloxacin (CIP 5) and Norfloxacin (NX 10), Aztreonam (AT 30), Gentamicin (GEN 10), Kanamycin (K 30), Azithromycin (AZM 30) Erythromycin (E 10), Imipenem (IMP 10), Chloramphenicol (C 30), Trimethoprim-Sulfamethoxazole (COT 25), Tetracycline (TE 30), Colistin (CL 10) and Polymyxin B (PB 300)).
(TIF)

**S7 Fig. Phenotypic identification of antibiotic resistant of *M. morganii and W. chitiniclastica* isolates found from layer chickens.** (Ampicillin (AMP25), Amoxicillin-clavulanic acid (AMC 30), Cefotaxime (CTX 30) Cefoxitin (CX 30) Ciprofloxacin (CIP 5) and Norfloxacin (NX 10), Aztreonam (AT 30), Gentamicin (GEN 10), Kanamycin (K 30), Azithromycin (AZM 30) Erythromycin (E 10), Imipenem (IMP 10), Chloramphenicol (C 30), Trimethoprim-Sulfamethoxazole (COT 25), Tetracycline (TE 30), Colistin (CL 10) and Polymyxin B (PB 300)).
(TIF)

**S8 Fig. Phenotypic identification of antibiotic resistant of *Aeromonas spp*. isolates found from layer chickens.** (Ampicillin (AMP25), Amoxicillin-clavulanic acid (AMC 30), Cefotaxime (CTX 30) Cefoxitin (CX 30) Ciprofloxacin (CIP 5) and Norfloxacin (NX 10), Aztreonam (AT 30), Gentamicin (GEN 10), Kanamycin (K 30), Azithromycin (AZM 30) Erythromycin (E 10), Imipenem (IMP 10), Chloramphenicol (C 30), Trimethoprim-Sulfamethoxazole (COT 25), Tetracycline (TE 30), Colistin (CL 10) and Polymyxin B (PB 300)).
(TIF)

**S1 Table. Characteristics of selected farms for sample collection.**
(DOCX)

**S2 Table. Characteristics of poultry chickens.**
(DOCX)

**S3 Table. Primers list for different genes used in this study.**
(DOCX)

**S4 Table. PCR mixture preparation for *rcsA* and *gyr-b* amplification.**
(DOCX)

**S5 Table. PCR condition for *rcsA* and *gyr-b* amplification.**
(DOCX)

**S6 Table. PCR mixture preparation for *16S* rRNA amplification.**
(DOCX)

**S7 Table. PCR Condition for *16S* rRNA amplification.**
(DOCX)

**S8 Table. PCR mixture preparation for ARG screening via PCR amplification.**
(DOCX)

**S9 Table. PCR condition for ESBL gene amplification.**
(DOCX)

**S10 Table. PCR condition for *tetA* and *tetB* amplification.**
(DOCX)

**S11 Table. PCR condition for *sul1* and *sul2* amplification.**
(DOCX)

**S12 Table. PCR condition for *mcr-1 gene* amplification.**
(DOCX)

**S13 Table. Standard Zone of Inhibition (ZOI) data to interpret antibiogram result of bacteria according to CLSI 2018 (8).**
(DOCX)

**S1 File. Sanger di-deoxy sequencing data.**
(DOCX)

## Author Contributions

**Conceptualization:** Md. Adnan Munim, Shipan Das Gupta.

**Data curation:** Md. Adnan Munim, Ithmam Hami.

**Formal analysis:** Md. Adnan Munim.

**Funding acquisition:** Shipan Das Gupta.

**Investigation:** Shipan Das Gupta.

**Methodology:** Md. Adnan Munim, Ithmam Hami, Mridul Gope Topu.

**Project administration:** Shipan Das Gupta.

**Resources:** Shuvo Chandra Das, Shipan Das Gupta.

**Supervision:** Shipan Das Gupta.

**Validation:** Ithmam Hami, Shipan Das Gupta.

**Visualization:** Md. Adnan Munim, Shuvo Chandra Das, Shipan Das Gupta.

**Writing – original draft:** Md. Adnan Munim.

**Writing – review & editing:** Shuvo Chandra Das, Md. Murad Hossain, Shipan Das Gupta.

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
