## [Editor Report · Decision Letter 0]

26 Oct 2023

PONE-D-23-31092Unveiling multi-drug resistant (MDR) gram negative pathogenic bacteria from poultry chickens in the Noakhali region of BangladeshPLOS ONE

Dear Dr. Gupta,

Thank you for submitting your manuscript to PLOS ONE. After careful consideration, we feel that it has merit but does not fully meet PLOS ONE’s publication criteria as it currently stands. Therefore, we invite you to submit a revised version of the manuscript that addresses the points raised during the review process.

We look forward to receiving your revised manuscript.

Kind regards,

Anselme Shyaka

Academic Editor

PLOS ONE

“This work was supported by the ‘Noakhali Science and Technology University- Research Cell’ Teachers’ grant of the budget year 2021-2022 (Project ID: NSTU/RC-BG-06/T-23/32).”

“This work was supported by the ‘Noakhali Science and Technology University- Research Cell’ Teachers’ grant of the budget year 2021-2022 (Project ID: NSTU/RC-BG-06/T-23/32). The authors thank to Biotechnology and Genetic Engineering department of NSTU for technical support.”

“This work was supported by the ‘Noakhali Science and Technology University- Research Cell’ Teachers’ grant of the budget year 2021-2022 (Project ID: NSTU/RC-BG-06/T-23/32).”

Additional Editor Comments:

Dear Author,

I would like to invite you to resubmit your manuscript after affecting the following modifications:

(1) Please resubmit Figure 1, as the legend is not readable.

(2) Ensure that you resubmit Figure 3 because it is not visible.

(3) Resubmit a clearer Figure 4A and 4B with legible legends.

(4) Upload a new Fig. 7 with a legible legend.

(5) Make sure that the table 2 rows are not breaking across pages (with text in between)

(6) Ensure that you remove the text highlight color used inside the manuscript.

Kindly note that your manuscript has not yet been assessed by reviewers.

I am looking forward to receiving your manuscript after the above changes.

---

## [Author Response · Author response to Decision Letter 0]

16 Nov 2023

#Editor’s comment 1: Please ensure that your manuscript meets PLOS ONE's style requirements, including those for file naming.

Authors’ Response: We express our gratitude to the editor for providing valuable guidance. In response to PLOS ONE's style requirements, we have now rectified the file names accordingly.

#Editor’s comment 2: Please state what role the funders took in the study. If the funders had no role, please state: "The funders had no role in study design, data collection and analysis, decision to publish, or preparation of the manuscript."

Authors’ Response: The funders contributed solely by providing financial support for this study. They were not involved in the study's design, data collection, analysis, the decision to publish, or the preparation of the manuscript.

#Editor’s comment 3: Please remove any funding-related text from the manuscript and let us know how you would like to update your Funding Statement.

Authors’ Response: Following the editor's guidance, we have removed the funding-related text from the manuscript. We kindly request that you update the funding statement as follows: 'This work received financial support from the 'Noakhali Science and Technology University-Research Cell' Teachers' grant for the budget year 2021-2022 (Project ID: NSTU/RC-BG-06/T-23/32). We appreciate your attention to this matter.

#Editor’s comment 4: We note that you have indicated that data from this study are available upon request. PLOS only allows data to be available upon request if there are legal or ethical restrictions on sharing data publicly. In your revised cover letter, please address the following prompts:

Authors’ Response: We express our gratitude to the editor for addressing this crucial matter. There are no ethical or legal restrictions preventing us from sharing a de-identified dataset. Therefore, we are pleased to share our raw dataset as a supplementary file. In the earlier version of the manuscript, a significant portion of the dataset was already included in the supplementary information. In the current version, we are uploading an Excel file (named 'Raw data') containing the most essential raw data information for this study. We believe that the information provided in the supplementary files is comprehensive enough to replicate the study

#Editor’s comment 5: PLOS ONE now requires that authors provide the original uncropped and unadjusted images underlying all blot or gel results reported in a submission’s figures or Supporting Information files. This policy and the journal’s other requirements for blot/gel reporting and figure preparation are described in detail at https://journals.plos.org/plosone/s/figures#loc-blot-and-gel-reporting-requirements and https://journals.plos.org/plosone/s/figures#loc-preparing-figures-from-image-files. When you submit your revised manuscript, please ensure that your figures adhere fully to these guidelines and provide the original underlying images for all blot or gel data reported in your submission. See the following link for instructions on providing the original image data: https://journals.plos.org/plosone/s/figures#loc-original-images-for-blots-and-gels.

Authors’ Response: In compliance with the PLOS ONE journal guidelines and following the editor's suggestion, we are pleased to share the unaltered gel images from this study. All the uncropped raw images have been consolidated in a single PDF file labeled ‘Raw gel image’

 #Editor’s comment 6: Please resubmit Figure 1, as the legend is not readable.

Authors’ Response: In the initial version of the manuscript, we included a figure with a resolution of 300 dpi, featuring a clearly legible legend. Unfortunately, during the PDF conversion for manuscript submission, the figure resolution was compromised, rendering the legend unreadable. We have now uploaded a new figure, although we cannot guarantee that the image will remain unaffected. In the event of a recurrence, we are willing to provide the image as an email attachment for clarity.

#Editor’s comment 7: Ensure that you resubmit Figure 3 because it is not visible.

Authors’ Response: Based on the editor's feedback, we have resubmitted Figure 3 with a higher resolution. We are optimistic that this revision will ensure the visibility of the figure in the submitted manuscript file.

#Editor’s comment 8: Resubmit a clearer Figure 4A and 4B with legible legends.

Authors’ Response: We have resubmitted Figure 4A and 4B with the correction of legible legends mentioned by editor.

#Editor’s comment 9: Upload a new Fig. 7 with a legible legend.

Authors’ Response: Due to the background color, the legends in Figure 7 were not visible. We have addressed this issue by replacing it with a resubmitted Figure 7 featuring a white background.

#Editor’s comment 10: Make sure that the table 2 rows are not breaking across pages (with text in between)

Authors’ Response: We express our gratitude to the editor for bringing this unintentional error to our attention. Subsequently, we have rectified Table 2, ensuring that the rows no longer break across the page.

#Editor’s comment 11: Ensure that you remove the text highlight color used inside the manuscript.

Authors’ Response: We have removed the text highlights from the revised manuscript.

---

## [Decision Letter · Decision Letter 1]

20 Mar 2024

PONE-D-23-31092R1Unveiling multi-drug resistant (MDR) gram negative pathogenic bacteria from poultry chickens in the Noakhali region of BangladeshPLOS ONE

Dear Dr. Gupta,

Thank you for submitting your manuscript to PLOS ONE. After careful consideration, we feel that it has merit but does not fully meet PLOS ONE’s publication criteria as it currently stands. Therefore, we invite you to submit a revised version of the manuscript that addresses the points raised during the review process.

We look forward to receiving your revised manuscript.

Kind regards,

Nabi Jomehzadeh, Ph.D (Assistant Professor)

Academic Editor

PLOS ONE

**Additional Editor Comments:**

 The article is very interesting and well-written; however, I believe it is very long, authors are advised to shorten it.

Reviewers' comments:

Reviewer's Responses to Questions

**Comments to the Author**

1. If the authors have adequately addressed your comments raised in a previous round of review and you feel that this manuscript is now acceptable for publication, you may indicate that here to bypass the “Comments to the Author” section, enter your conflict of interest statement in the “Confidential to Editor” section, and submit your "Accept" recommendation.

Reviewer #1: All comments have been addressed

Reviewer #2: (No Response)

2. Is the manuscript technically sound, and do the data support the conclusions?

Reviewer #1: Yes

Reviewer #2: Yes

3. Has the statistical analysis been performed appropriately and rigorously? 

Reviewer #1: Yes

Reviewer #2: Yes

4. Have the authors made all data underlying the findings in their manuscript fully available?

Reviewer #1: Yes

Reviewer #2: Yes

5. Is the manuscript presented in an intelligible fashion and written in standard English?

Reviewer #1: Yes

Reviewer #2: Yes

6. Review Comments to the Author

Reviewer #1: Line 1-2. The “Unveiling” in the title sounds a bit inappropriate and therefore suggests that the title reads “Multi-drug resistant (MDR) gram negative pathogenic bacteria isolated from poultry in the Noakhali region of Bangladesh”. Poultry chicken sounds repetitive.

Line 38-39 : “is yet to understand” must read “ is yet to be understood”

Line 30-31 : I suggest “ESBL” and other first mentions must be appropriately defined

Line 53 : Now-a-days must be replaced with “Nowadays” or better still, “More recently”

Line 66-68 : “Try to rephrase this statement. Perhaps the sentence will read good if " all kinds of people are benefited" is replaced with " many people benefit”

Line 75-76 : Consider changing serovars to spp. and for uniformity, stick to spp.

Line 163: Just for clarification. Is this supposed to be 1ul or 100ul or 1ml as 1µl currently reported seems so small for centrifuging and pipetting off supernatant.

Line 175: “S1, S2 and S3 Tables“ should be better placed as : “Supplementary data S1, S2 and S3 Tables”

Line 195: The referencing of “Hudzicki” must be done well.

Line 239-247: The “Selection and criteria of study site” must be incorporated into the methodology and not results as it presents no results.

Line 399-400: The sentence does not read right and rephrasing will do.

Line 413-414 : Fig. 7 is good. However, if a table showing the various measured of the zones of inhibition as against the standards, it would have been great. This link can provide the reference point for number of antimicrobials : https://www.cdc.gov/narms/antibiotics-tested.html.

Line 422: “Poultry” must be replaced with “Poultry production” to make the sentence read better.

Line 454 : “To our best known” should read “ "To the best of our knowledge" to sound better.

Line 471: “In compare” should read “ In comparison.”

The article is good however, lots of sentence reconstruction needs to be done. The authors may want to use some English Language proof-reading aid them in improving on the current nature of the manuscript.

Reviewer #2: 1- In title and all the manuscript, write ‘Gram’ not ‘gram’: Unveiling multi-drug resistant (MDR) Gram-negative pathogenic bacteria from poultry chickens in the Noakhali region of Bangladesh.

2- Lin 53, write :’... antibiotic resistant (AR) bacteria....’

3- In line 61, you provide the quantity of used antibiotics (8,164,662 kg); however, the reference is since 2009 (Dhanarani TS, Shankar C, Park J, Dexilin M, Kumar RR, Thamaraiselvi K. Study on acquisition of bacterial antibiotic resistance determinants in poultry litter. Poultry science. 2009;88(7):1381-7), this is a very ancient data. Please provide more recent data, now we are in 2024, certainly the quantity of antibiotics used was changed.

4- The term ‘Enterobacteriaceae’ must be written in italic, verif this in all the manuscript.

5- Correct as follow: ‘Enterobacteriaceae frequently serve as carriers of extended-spectrum β-lactamase (ESBL)-encoding genes and are frequently........’

6- Correct as follows: ‘..The presence of blaCTX-M, blaSHV, and blaTEM genes encoding CTX-M, TEM, and SHV β-lactamases, respectively, empowers these bacteria with resistance against penicillins, first, second, and third-generation cephalosporins, as well as aztreonam (17, 18).’ PLEASE ‘bla’ must be in italic and the name of the enzyme as indice (under ‘bla’) and not in italic.

7- In the sentence ‘Notably, studies by Kluytmans et al. and Leverstein van Hall et al. have highlighted strong genetic similarities between ESBL-producing E. coli 90 isolated from chicken meat and humans’ Please add the number of the references ‘Kluytmans et al. (its number in the list of reference)’ and ‘Leverstein van Hall et al (its number in the list of reference).

8- Line 163, in DNA extraction, I believe that you centrifuged 1 ml of culture not just 1 µl; please verify and correct this. IN ADDITION, just a remark, 30 min of heating is a lot, 10min to 12 min at 100 °C is enough to get the lyses of bacteria, especially Gram-negative bacteria with fragile peptidoglycan wall.

9- DNA extraction, please delete the step ‘x’; you already collected supernatant that contain DNA for PCR, why you heat it again??? This is incorrect, in this step ‘x’ you will make DNA in two strands, why do it?, this step can degrade DNA, which false PCR.

10- Line 195, please correct it is ‘2009’ not ‘20090’ (.....Kirby-Bauer disk-diffusion method according to Hudzicki, (20090 (43).’ I do not know if life still to that dates 20090.

11- In antimicrobial susceptibility test you divided beta-lactams to : ‘Ampicillin (AMP25) and Amoxicillin-clavulanic acid (AMC 30) from β-lactams ; Cefotaxime 201 (CTX 30) and Cefoxitin (CX 30) from Cephems; Aztreonam (AT 30) from monobactams; Imipenem (IMP 10) from carbapenems. THIS classification is somewhat correct. However, these antibiotics must be considered as one group (Beta-lactams) when you define MDR. The MDR is for strains showing resistance to three or more families (Beta-lactams are: Ampicillin, Amoxicillin-clavulanic acid; Cefotaxime, Cefoxitin; Aztreonam; Imipenem), so be careful when defining MDR strains.

12- Line 226, tou wrote ‘...alongside the New Delhi Metallo β-lactamase (blaNDM) gene’. PLEASE INDICATE THAT IT CODE A metallo-carbapenemase to avoid misunderstanding as an ESBL gene; NDM is not an ESBL.

13- Line 229, you wrote ‘...while sul1 and sul2 were examined for Trimethoprim-Sulfamethoxazole resistance’. In reality this is not perfectly correct. The sul 1, sul 2 and sul3 encode resistance to sulfonamides not to trimethoprim. Yes when the strain is resistant to sulfamethoxazole it can be also resistant to trimethoprim since the biochemical process of folates synthesis is interrupted. Trimethoprime-resistance is encoded by dhfr genes.

14- Line 230, modify the sentence as follow:’Moreover, among the globally described ten mcr variants encoding colistin resistance, only mcr-1 gene was investigated by PCR since it is more prevalent worldwide.’

15- Figure 1 (A and B), please correct ‘K. pneumoniae’ not ‘K. pneumonia’. Correct this also in Figure 2 and Table 1 and table 2, verify this in all the manuscript. Please correct ‘gyr B’ not ‘gyr-b’ in figure 2.

16- PLEASE IN THE TEXT PROVIDE THE NUMBER OF ISOLATES OF EACH IDENTIFIED SPECIES. I want to know the number of E. coli, K. pneumoniae...... Please provide the number of species and the percentages (from broiler and from layer; and from the total). For example, after reading the article I do not know the number of E. coli isolates!!!!.

17- WHEN you provide any data about antimicrobial resistance or genes please provide the number and the frequency (n (%)).

18- Line 289, write: ‘Phylogenetic analysis of identified bacterial isolates’

19- In section ‘results’, ‘Phenotypic characterization of antibiotic resistant pattern’ need revision. Please try to present percentages of resistance in simple form. Just few sentences can explain the detected resistance. The paragraph is very long, try to shorten it by presenting simple and clear results. Readers cannot follow what you wrote. PLEASE COMAPARISON OR CONCLUSION OR DISCUSSION OF YOUR RESULTS MUST BE DONE IN SECTION ‘DISCUSSION’

20- In section Results, sentences should be short, clear; especially you provided a lot of tables, figures and supplementary files, sometimes are enough. Paragraphs are in bloc, try to subdivide them. Readers cannot read paragraph containing 20 lines, when you cut the long paragraph to 3 paragraphs it will be better

21- Change the title of table 3, as follows: Table 3: Antimicrobial susceptibilities of Gram-negative isolates collected from broiler and layer chicken samples.

22- In table 3, I believe that it is not correct to make the sum of all Gram negative bacteria. In your collection there are some species that are naturally resistant to some antibiotics. I advise to delete this table since in table 2 you perfectly provided the resistance profile of each isolate of each species, this is enough. Really I disliked this table 3.

23- For example this paragraph can be as follows (I make few modification also):

Upon evaluating ESBL gene presence and other genes encoding resistance markers in collected isolates, we observed that lactose-positive and lactose-negative E. coli carried blaTEM, tet A, tet B, sul 1, and sul 2 genes, with the latter also harboring blaSHV (Figs 5A, 5B, 5C). These E. coli isolates displayed resistance to β-lactam combination agents, penicillin, cephalosporins, tetracycline, and folate pathway antagonists. Similarly, K. pneumoniae isolates from broiler and layer chickens showed the presence of blaTEM, blaSHV, sul 1, and tet A genes (Fig 5D).

A significant portion of K. pneumoniae isolates from both groups exhibited resistance to β-lactam combination agents, penicillin, cephalosporins, tetracycline, and folate pathway antagonists. Among the isolates, E. hormaechei, P. penneri, P. stuartii, W. chitiniclastica, and M. morganii carried tet A, sul 1, blaSHV, and blaTEM genes (Figs 5E-G, 5I-J) aligning with their phenotypic resistance.

Lastly, S. enterica isolates carried sul 1, sul 2, blaTEM genes (Fig 5H) and most of the S. enterica isolates shown resistance to β-lactam combination agents, penicillin, cephalosporins, tetracycline, and folate pathway antagonists. As those isolates showed resistance to tetracycline without the presence of both tet A and tet B genes indicated that the presence of other genes encoding tetracycline resistance, which were not investigated in this study, in S. enterica isolates.’

24- It is not correct to say about genes ‘isoforms’ or ‘isotype’, in biochemistry this is correct but in genetic (gene) this is not correct. You wrote in line 397 ‘mcr1isotype’, this can be wrote in this case as ‘mcr-1 variant’.

25- Please verify the order of figures and tables.

26- The mcr1 must be written in all the text and figures and tables as ‘mcr-1’, this is the international nomenclature of mcr genes.

27- Discussion is very long try to shorten it.

28- PLEASE IN SECTION DISCUSSION NEVER REFER TO TABLES AND FIGURES, THIS WAS ALREADY DONE IN SECTION RESULTS.

7. PLOS authors have the option to publish the peer review history of their article (what does this mean?). If published, this will include your full peer review and any attached files.

Reviewer #1: No

Reviewer #2: No

---

## [Author Response · Author response to Decision Letter 1]

18 Apr 2024

Dear Editor,

Thank you very much for allowing us the opportunity to submit the revised version of our manuscript entitled “Unveiling multi-drug resistant (MDR) Gram-negative pathogenic bacteria from poultry chickens in the Noakhali region of Bangladesh” for publication in the ‘Plos One'. We really appreciate the time and that you and the reviewers dedicated to providing feedback on our manuscript and are grateful for the insightful comments on and valuable improvements to our paper. We have incorporated all the suggestions made by the reviewers. Please see below, in blue, for a point-by-point response to the reviewers’ comments and concerns.

Reviewer #1:

#Comment 1: Line 1-2. The “Unveiling” in the title sounds a bit inappropriate and therefore suggests that the title reads “Multi-drug resistant (MDR) gram negative pathogenic bacteria isolated from poultry in the Noakhali region of Bangladesh”. Poultry chicken sounds repetitive.

Author’s response: We express our gratitude to the reviewer for providing the valuable guidance. In response to reviewer suggestion, we have changed the title in the revised manuscript.

#Comment 2: Line 38-39: “is yet to understand” must read “is yet to be understood”.

Author’s response: We have changed in the revised manuscript.

#Comment 3: Line 30-31: I suggest “ESBL”, and other first mentions must be appropriately defined

Author’s response: According to the reviewer suggestion we have added the abbreviations in the revised manuscript.

#Comment 4: Line 53: Now-a-days must be replaced with “Nowadays” or better still, “More recently”

Author’s response: We have replaced the word in the revised manuscript.

#Comment 5: “Try to rephrase this statement. Perhaps the sentence will read good if " all kinds of people are benefited" is replaced with " many people benefit”.

Author’s response: We have rephrased the sentence as per reviewer’s suggestion.

#Comment 6: Consider changing serovars to spp. and for uniformity, stick to spp.

Author’s response: In the revised manuscript, we have changed all serovars to spp. according to the reviewer’s advice.

#Comment 7: Just for clarification. Is this supposed to be 1ul or 100ul or 1ml as 1µl currently reported seems so small for centrifuging and pipetting off supernatant.

Author’s response: We are grateful to the reviewer for identifying this unintentional error. Thanks to their diligence, we have now corrected the amount of broth culture required for DNA extraction in the revised manuscript.

#Comment 8: “S1, S2 and S3 Tables” should be better placed as: “Supplementary data S1, S2 and S3 Tables”

Author’s response: We have corrected the data and table information in the revised manuscript.

#Comment 9: Line 195: The referencing of “Hudzicki” must be done well.

Author’s Response: We are thankful to the reviewer for his/her prudent suggestions. According to suggestions we have made the changes in the revised manuscript. 

Comment 10: The “Selection and criteria of study site” must be incorporated into the methodology and not results as it presents no results.

Author’s Response: In response to the reviewer's feedback, we have replaced the 'Selection and Criteria of Study Site' section within the method section of the revised manuscript.

Comment 11: Line 399-400: The sentence does not read right, and rephrasing will do.

Author’s response: we have rephrased the sentence as reviewer suggested.

Comment 12: Line 413-414: Fig. 7 is good. However, if a table showing the various measured of the zones of inhibition as against the standards, it would have been great. This link can provide the reference point for number of antimicrobials: https://www.cdc.gov/narms/antibiotics-tested.html.

Author’s response: We extend our gratitude to the reviewer for their invaluable advice. Following their guidance, we have included additional supplementary tables (S13) detailing the measured zone of inhibition (ZOI) for each antibiotic. Additionally, we have integrated this ZOI information into Figure 7 

Comment 13: Line 422: “Poultry” must be replaced with “Poultry production” to make the sentence read better.

Author’s response: We have replaced the word in the revised manuscript.

Comment 14: Line 454: “To our best known” should read “To the best of our knowledge" to sound better.

Author’s response: We have made the change in the revised manuscript.

Comment 15: Line 471: “In compare” should read “In comparison.”

Author’s response: We have replaced the words as reviewer suggested in the revised manuscript. 

Reviewer #2:

Comment 1: In title and all the manuscript, write ‘Gram’ not ‘gram’: Unveiling multi-drug resistant (MDR) Gram-negative pathogenic bacteria from poultry chickens in the Noakhali region of Bangladesh.

Author’s response: We are grateful to the reviewer for his suggestion. We have corrected the word in the revised manuscript.

Comment 2: Lin 53, write :’... antibiotic resistant (AR) bacteria....’

Author’s response: we have added the word as reviewer suggested.

Comment 3: In line 61, you provide the quantity of used antibiotics (8,164,662 kg); however, the reference is since 2009 (Dhanarani TS, Shankar C, Park J, Dexilin M, Kumar RR, Thamaraiselvi K. Study on acquisition of bacterial antibiotic resistance determinants in poultry litter. Poultry science. 2009;88(7):1381-7), this is a very ancient data. Please provide more recent data, now we are in 2024, certainly the quantity of antibiotics used was changed.

Author’s response: We have added the recent data with citation as reviewer suggested in the revised manuscript.

Comment 4: The term ‘Enterobacteriaceae’ must be written in italic, verif this in all the manuscript.

Author’s response: We have ensured that all instances of the term 'Enterobacteriaceae' are correctly formatted in italics throughout the revised manuscript

Comment 5: Correct as follow: ‘Enterobacteriaceae frequently serve as carriers of extended-spectrum β-lactamase (ESBL)-encoding genes and are frequently........’

Author’s response: We are thankful to the reviewer for the suggestion and corrected in the revised manuscript.

Comment 6: Correct as follows: ‘..The presence of blaCTX-M, blaSHV, and blaTEM genes encoding CTX-M, TEM, and SHV β-lactamases, respectively, empowers these bacteria with resistance against penicillins, first, second, and third-generation cephalosporins, as well as aztreonam (17, 18).’ PLEASE ‘bla’ must be in italic and the name of the enzyme as indice (under ‘bla’) and not in italic.

Author’s response: We extend our gratitude to the reviewer for their invaluable feedback. As per the reviewer's advice, we have implemented the necessary corrections.

Comment 7: In the sentence ‘Notably, studies by Kluytmans et al. and Leverstein van Hall et al. have highlighted strong genetic similarities between ESBL-producing E. coli 90 isolated from chicken meat and humans’ Please add the number of the references ‘Kluytmans et al. (its number in the list of reference)’ and ‘Leverstein van Hall et al (its number in the list of reference).

Author’s response: We are grateful to the reviewer for identifying our unintentional error. We have promptly rectified this oversight by adding the reference to the list

Comment 8: Line 163, in DNA extraction, I believe that you centrifuged 1 ml of culture not just 1 µl; please verify and correct this. IN ADDITION, just a remark, 30 min of heating is a lot, 10min to 12 min at 100 °C is enough to get the lyses of bacteria, especially Gram-negative bacteria with fragile peptidoglycan wall.

Author’s response: We appreciate the reviewer for identifying our typo mistake. We have corrected the measurement from 1 µl culture to 1 ml of culture. Additionally, we thank reviewer for his insightful remarks. Going forward, we will explore reduced heating times for DNA extraction from Gram-negative bacteria

Comment 9: DNA extraction, please delete the step ‘x’; you already collected supernatant that contain DNA for PCR, why you heat it again??? This is incorrect, in this step ‘x’ you will make DNA in two strands, why do it?, this step can degrade DNA, which false PCR.

Author’s response: In accordance with the reviewer's suggestion, we have removed step 'x' from the DNA extraction protocol. Employing this revised protocol, we successfully extracted DNA from Gram-negative bacteria for another project. We appreciate the valuable advice, which not only streamlined our protocol but also reduced the DNA extraction time. Thank you for your guidance.

Comment 10: Line 195, please correct it is ‘2009’ not ‘20090’ (.....Kirby-Bauer disk-diffusion method according to Hudzicki, (20090 (43).’ I do not know if life still to that dates 20090.

Author’s response: We have now corrected the typo. Thank you for bringing it to our attention.

Comment 11: In antimicrobial susceptibility test you divided beta-lactams to : ‘Ampicillin (AMP25) and Amoxicillin-clavulanic acid (AMC 30) from β-lactams ; Cefotaxime 201 (CTX 30) and Cefoxitin (CX 30) from Cephems; Aztreonam (AT 30) from monobactams; Imipenem (IMP 10) from carbapenems. THIS classification is somewhat correct. However, these antibiotics must be considered as one group (Beta-lactams) when you define MDR. The MDR is for strains showing resistance to three or more families (Beta-lactams are: Ampicillin, Amoxicillin-clavulanic acid; Cefotaxime, Cefoxitin; Aztreonam; Imipenem), so be careful when defining MDR strains.

Author’s response: We express our gratitude to the reviewer for their insightful feedback. We have diligently incorporated the suggested correction into the revised manuscript.

Comment 12: Line 226, tou wrote ‘...alongside the New Delhi Metallo β-lactamase (blaNDM) gene’. PLEASE INDICATE THAT IT CODE A metallo-carbapenemase to avoid misunderstanding as an ESBL gene; NDM is not an ESBL.

Author’s response: We have revised the sentence based on the reviewer's advice

Comment 13: Line 229, you wrote ‘...while sul1 and sul2 were examined for Trimethoprim-Sulfamethoxazole resistance’. In reality this is not perfectly correct. The sul 1, sul 2 and sul3 encode resistance to sulfonamides not to trimethoprim. Yes when the strain is resistant to sulfamethoxazole it can be also resistant to trimethoprim since the biochemical process of folates synthesis is interrupted. Trimethoprime-resistance is encoded by dhfr genes.

Author’s response: We sincerely appreciate the invaluable advice provided by the reviewer. Following their guidance, we have now rectified the sentence accordingly in the revised manuscript

Comment 14: Line 230, modify the sentence as follow:’Moreover, among the globally described ten mcr variants encoding colistin resistance, only mcr-1 gene was investigated by PCR since it is more prevalent worldwide.’

Author’s response: We extend our gratitude to reviewer for improving the sentence.

Comment 15: Figure 1 (A and B), please correct ‘K. pneumoniae’ not ‘K. pneumonia’. Correct this also in Figure 2 and Table 1 and table 2, verify this in all the manuscript. Please correct ‘gyr B’ not ‘gyr-b’ in figure 2.

Author’s response: We have corrected the following suggestions made by reviewer in the revised manuscript. 

Comment 16: PLEASE IN THE TEXT PROVIDE THE NUMBER OF ISOLATES OF EACH IDENTIFIED SPECIES. I want to know the number of E. coli, K. pneumoniae...... Please provide the number of species and the percentages (from broiler and from layer; and from the total). For example, after reading the article I do not know the number of E. coli isolates!!!!.

Author’s response: We have added the number of isolates of each identified species in the revised manuscript. 

Comment 17: WHEN you provide any data about antimicrobial resistance or genes please provide the number and the frequency (n (%)).

Author’s response: With all due respect to the reviewer, we would like to offer clarification regarding why we did not present the number and frequency of analyzed antimicrobial genes. In our antibiogram test, we meticulously displayed the comprehensive antibiotic resistance profiles of all isolates for each species (refer to Table 2). However, for the purpose of DNA extraction and MDR gene detection via PCR, we deliberately selected only the top three isolates from each species based on their highest MAR index. Our intention was to establish a correlation between the phenotypic antibiotic resistance results and the PCR-based detection of antibiotic-resistant genes. Hence, the omission of the specific number and frequency of antibiotic-resistant genes was motivated by our focus on this correlation

Comment 18: Line 289, write: ‘Phylogenetic analysis of identified bacterial isolates’

Author’s response: We have changed this according to the reviewer suggestion.

Comment 19: In section ‘results’, ‘Phenotypic characterization of antibiotic resistant pattern’ need revision. Please try to present percentages of resistance in simple form. Just few sentences can explain the detected resistance. The paragraph is very long, try to shorten it by presenting simple and clear results. Readers cannot follow what you wrote. PLEASE COMAPARISON OR CONCLUSION OR DISCUSSION OF YOUR RESULTS MUST BE DONE IN SECTION ‘DISCUSSION’

Author’s response: We extend our gratitude to the reviewer for their invaluable advice. We concur with the reviewer's observation regarding the size of this section. In response to previous suggestions, such as incorporating isolate numbers and percentages, we have made essential revisions. Additionally, in line with the reviewer's guidance, we have endeavored to streamline this section, focusing solely on the analysis findings. We anticipate that these revisions will ease readers to follow the story of the study.

Comment 20: In section Results, sentences should be short, clear; especially you provided a lot of tables, figures and supplementary files, sometimes are enough. Paragraphs are in bloc, try to subdivide them. Readers cannot read paragraph containing 20 lines, when you cut the long paragraph to 3 paragraphs it will be better

Author’s response: In response to the reviewer's advice and valuable feedback, we have implemented necessary corrections in the results section. While ensuring the original narrative remains intact, we have removed several sentences, resulting in a slightly shorter revised section compared to the previous version.

Comment 21: Change the title of table 3, as follows: Table 3: Antimicrobial susceptibilities of Gram-negative isolates collected from broiler and layer chicken samples.

Author’s response: We have removed the table 3 according to reviewer suggestion.

Comment 22: In table 3, I believe that it is not correct to make the sum of all Gram negative bacteria. In your collection there are some species that are naturally resistant to some antibiotics. I advise to delete this table since in table 2 you perfectly provided the resistance profile of each isolate of each species, this is enough. Really I disliked this table 3.

Author’s response: We have removed the table 3 according to reviewer suggestion.

Comment 23: For example this paragraph can be as follows (I make few modification also):

Upon evaluating ESBL gene presence and other genes encoding resistance markers in collected isolates, we observed that lactose-positive and lactose-negative E. coli carried blaTEM, tet A, tet B, sul 1, and sul 2 genes, with the latter also harboring blaSHV (Figs 5A, 5B, 5C). These E. coli isolates displayed resistance to β-lactam combination agents, penicillin, cephalosporins, tetracycline, and folate pathway antagonists. Similarly, K. pneumoniae isolates from broiler and layer chickens showed the presence of blaTEM, blaSHV, sul 1, and tet A genes (Fig 5D).

A significant portion of K. pneumoniae isolates from both groups exhibited resistance to β-lactam combination agents, penicillin, cephalosporins, tetracycline, and folate pathway antagonists. Among the isolates, E. hormaechei, P. penneri, P. stuartii, W. chitiniclastica, and M. morganii carried tet A, sul 1, blaSHV, and blaTEM genes (Figs 5E-G, 5I-J) aligning with their phenotypic resistance.

Lastly, S. enterica isolates carried sul 1, sul 2, blaTEM genes (Fig 5H) and most of the S. enterica isolates shown resistance to β-lactam combination agents, penicillin, cephalosporins, 

---

## [Decision Letter · Decision Letter 2]

17 May 2024

PONE-D-23-31092R2Multi-drug resistant (MDR) Gram-negative pathogenic bacteria from poultry in the Noakhali region of BangladeshPLOS ONE

Dear Dr. Gupta,

Thank you for submitting your manuscript to PLOS ONE. After careful consideration, we feel that it has merit but does not fully meet PLOS ONE’s publication criteria as it currently stands. Therefore, we invite you to submit a revised version of the manuscript that addresses the points raised during the review process.

We look forward to receiving your revised manuscript.

Kind regards,

Nabi Jomehzadeh, Ph.D (Assistant Professor)

Academic Editor

PLOS ONE

Journal Requirements:

Additional Editor Comments:

Dear Author,

I would like to invite you to resubmit your manuscript after affecting the following modifications:

(1) Please resubmit Figure 1, as the legend is not readable.

(2) Ensure that you resubmit Figure 3 because it is not visible.

(3) Resubmit a clearer Figure 4A and 4B with legible legends.

(4) Upload a new Fig. 7 with a legible legend.

(5) Make sure that the table 2 rows are not breaking across pages (with text in between)

(6) Ensure that you remove the text highlight color used inside the manuscript.

I am looking forward to receiving your manuscript after the above changes.

Reviewers' comments:

Reviewer's Responses to Questions

**Comments to the Author**

1. If the authors have adequately addressed your comments raised in a previous round of review and you feel that this manuscript is now acceptable for publication, you may indicate that here to bypass the “Comments to the Author” section, enter your conflict of interest statement in the “Confidential to Editor” section, and submit your "Accept" recommendation.

Reviewer #1: (No Response)

Reviewer #2: All comments have been addressed

2. Is the manuscript technically sound, and do the data support the conclusions?

Reviewer #1: No

Reviewer #2: Yes

3. Has the statistical analysis been performed appropriately and rigorously? 

Reviewer #1: I Don't Know

Reviewer #2: Yes

4. Have the authors made all data underlying the findings in their manuscript fully available?

Reviewer #1: Yes

Reviewer #2: Yes

5. Is the manuscript presented in an intelligible fashion and written in standard English?

Reviewer #1: No

Reviewer #2: Yes

6. Review Comments to the Author

Reviewer #1: The authors should be more diligent in their write up and be more coherent as the whole article is riffed with grammatical errors.

Line 1: The title will read better if written as “Multi-drug resistant (MDR) Gram-negative pathogenic bacteria isolated from poultry in the Noakhali region of Bangladesh”

Line 22: Please change all multi drug to “multi-drug”

Line 30: “Targeted gene sequences were amplified for detection ….” Should read “Targeted gene sequences were amplified for the detection….

Line 150: What do authors mean by “distinct chickens”?

Line 159: Authors should state the quantity of Peptone water broth used and better still if the methodology was adopted it must be referenced unless otherwise

Line 163: Full meaning of XLD is “Xylose Lysine Deoxycholate” and not what the authors have stated.

Line 173: The step iii as defined by the authors is unclear. Does this imply that after removal of the supernatant, the same tube is again filled with 1ml of pre-enrichment culture and centrifuged again?

Line 193: There is an excess single space after “blastn”

Line 199: Authors must provide the online link to this MAFFT tool and appropriately provide a citation

Line 199: Authors must be sure MAFFT was used for building phylogenies as the tool is mostly used for building sequence alignments from which phylogenies are built using tools like ClustaW

Lines 209-212: There are a series of “;;” which are inappropriate

Line 285: There is a repetition of “isolates”

These are only but a few comments as I could not write out all I could find

Reviewer #2: I revised carefully this manuscript, it seems better than the first submission especially that authors have also considered the revision advised by the reviewer 1. Authors have responded perfectly to my previous comments. Therefore, I have not other comments.

Sincerely

7. PLOS authors have the option to publish the peer review history of their article (what does this mean?). If published, this will include your full peer review and any attached files.

Reviewer #1: No

Reviewer #2: **Yes: **Mohamed Salah Abbassi

---

## [Author Response · Author response to Decision Letter 2]

20 May 2024

#Editor

#Editor’s comment 1: Please resubmit Figure 1, as the legend is not readable.

Author’s response: In the initial version of the manuscript, we included a figure with a resolution of 300 dpi and a clearly legible legend. Unfortunately, during the PDF conversion for manuscript submission, the figure resolution was compromised, making the legend unreadable. We have now uploaded a new figure. Additionally, we have attached Figure 1 to the email in case the image quality is compromised in the online version.

#Editor’s comment 2: Ensure that you resubmit Figure 3 because it is not visible.

Author’s response: Based on the editor's feedback, we have resubmitted Figure 3 with a higher resolution. We are optimistic that this revision will ensure the visibility of the figure in the submitted manuscript file. Additionally, we have submitted the Figure 3 in the E-mail attachment. 

#Editor’s comment 3: Resubmit a clearer Figure 4A and 4B with legible legends.

Author’s response: We have resubmitted Figure 4A and 4B with the correction of legible legends mentioned by editor. Additionally, we have submitted the Figure 4A and 4B in the E-mail attachment

#Editor’s comment 4: Upload a new Fig. 7 with a legible legend.

Author’s response: We have uploaded a new Figure 7 according to the editor’s advice. Additionally, we have attached Figure 7 to the email in case the image quality is compromised in the online version.

#Editor’s comment 5: Make sure that the table 2 rows are not breaking across pages (with text in between)

Author’s response: We have rectified Table 2, ensuring that the rows no longer break across the page.

#Editor’s comment 6: Ensure that you remove the text highlight color used inside the manuscript.

Author’s response: We have thoroughly checked and ensured that there is no text highlight in the revised manuscript

Reviewer #1:

The authors should be more diligent in their write up and be more coherent as the whole article is riffed with grammatical errors.

Author’s response: We appreciate the reviewer’s valuable feedback. We have thoroughly checked the manuscript for grammatical errors with the help of a native English speaker and identified some errors, which have been corrected in the revised version. Additionally, some sentences have been improved, and these changes can be easily seen in the track change version of the revised manuscript.

#Comment 1: Line 1: The title will read better if written as “Multi-drug resistant (MDR) Gram-negative pathogenic bacteria isolated from poultry in the Noakhali region of Bangladesh”

Author’s response: We express our gratitude to the Reviewer for his valuable feedback. According to the reviewer’s suggestion we have changed the title in the revised manuscript. 

#Comment 2: Line 22: Please change all multi drug to “multi-drug”

Author’s response: In the revised manuscript we have changed all the ‘multi drug’ to ‘multi-drug’.

#Comment 3: Line 30: “Targeted gene sequences were amplified for detection ….” Should read “Targeted gene sequences were amplified for the detection….

Author’s response: We thanks reviewer for his guidance. We have made the necessary change according to reviewer advice.

#Comment 4: Line 150: What do authors mean by “distinct chickens”?

Author’s response: In the revised manuscript, we have eliminated the word "distinct" from the line.

#Comment 5: Line 159: Authors should state the quantity of Peptone water broth used and better still if the methodology was adopted it must be referenced unless otherwise

Author’s response: In accordance with the reviewer's suggestion, we have added the quantity of peptone water broth used for the culture, along with the appropriate citation

#Comment 6: Line 163: Full meaning of XLD is “Xylose Lysine Deoxycholate” and not what the authors have stated.

Author’s response: We appreciate the reviewer for identifying our typographical error. We have corrected the full meaning of XLD in the revised manuscript.

#Comment 7: Line 173: The step iii as defined by the authors is unclear. Does this imply that after removal of the supernatant, the same tube is again filled with 1ml of pre-enrichment culture and centrifuged again?

Author’s response: We appreciate the reviewer’s query. Yes, after removing the supernatant, the same tube is refilled with 1 ml of pre-enrichment culture and centrifuged again to achieve a higher concentration of bacterial cells.

#Comment 8: Line 193: There is an excess single space after “blastn”

Author’s response: We thanks reviewer for bringing this unintentional error to our attention. We remove the excess single space in the revised manuscript. 

#Comment 9: Line 199: Authors must provide the online link to this MAFFT tool and appropriately provide a citation

Author’s response: We appreciate the reviewer's valuable advice. Following his suggestion, we have included an online link to the MAFFT tool for phylogenetic tree construction, along with the appropriate citation.

#Comment 9: Line 199: Authors must be sure MAFFT was used for building phylogenies as the tool is mostly used for building sequence alignments from which phylogenies are built using tools like ClustaW

Author’s response: We have ensured that the MAFFT tool was used for building the phylogenetic tree. Additionally, we have cited relevant articles (references 43 and 44) for MAFFT-based phylogenetic analysis.

#Comment 10: Lines 209-212: There are a series of “;;” which are inappropriate

Author’s response: We thank the reviewer for identifying the typos. We have removed the extra semicolon in the revised manuscript.

#Comment 11: Line 285: There is a repetition of “isolates”

Author’s response: The redundant word “isolates” has been removed in the revised manuscript.

---

## [Editor Report · Decision Letter 3]

22 May 2024

Multi-drug resistant (MDR) Gram-negative pathogenic bacteria isolated from poultry in the Noakhali region of Bangladesh

PONE-D-23-31092R3

Dear Dr. Gupta,

We’re pleased to inform you that your manuscript has been judged scientifically suitable for publication and will be formally accepted for publication once it meets all outstanding technical requirements.

Kind regards,

Nabi Jomehzadeh, Ph.D (Assistant Professor)

Academic Editor

PLOS ONE
---

## [Editor Report · Acceptance letter]

27 Jun 2024

PONE-D-23-31092R3 

PLOS ONE

Dear Dr. Gupta, 

I'm pleased to inform you that your manuscript has been deemed suitable for publication in PLOS ONE. Congratulations! Your manuscript is now being handed over to our production team.

Kind regards, 

on behalf of

Dr. Nabi Jomehzadeh 

Academic Editor

PLOS ONE